# Notochord vacuoles absorb compressive bone growth during zebrafish spine formation

Jennifer Bagwell[1], James Norman[1], Kathryn Ellis[1†], Brianna Peskin[1], James Hwang[1], Xiaoyan Ge[2‡], Stacy V Nguyen[3], Sarah K McMenamin[3], Didier YR Stainier[2,4], Michel Bagnat[1*]

[1]Department of Cell Biology, Duke University, Durham, United States; [2]Department of Biochemistry & Biophysics, University of California, San Francisco, San Francisco, United States; [3]Biology Department, Boston College, Boston, United States; [4]Max Planck Institute for Heart and Lung Research, Bad Nauheim, Germany

*For correspondence:
michel.bagnat@duke.edu

Present address: †Decibel Therapeutics, Boston, United States; ‡Icahn School of Medicine at Mount Sinai, New York, United States

**Abstract** The vertebral column or spine assembles around the notochord rod which contains a core made of large vacuolated cells. Each vacuolated cell possesses a single fluid-filled vacuole, and loss or fragmentation of these vacuoles in zebrafish leads to spine kinking. Here, we identified a mutation in the kinase gene *dstyk* that causes fragmentation of notochord vacuoles and a severe congenital scoliosis-like phenotype in zebrafish. Live imaging revealed that Dstyk regulates fusion of membranes with the vacuole. We find that localized disruption of notochord vacuoles causes vertebral malformation and curving of the spine axis at those sites. Accordingly, in *dstyk* mutants the spine curves increasingly over time as vertebral bone formation compresses the notochord asymmetrically, causing vertebral malformations and kinking of the axis. Together, our data show that notochord vacuoles function as a hydrostatic scaffold that guides symmetrical growth of vertebrae and spine formation.

## Introduction

The notochord is the main structural element of the anteroposterior (AP) body axis in chordates, and in vertebrates it also serves as a scaffold for spine formation. In basal chordates such as ascidians, the notochord is a simple and continuous fluid filled tube that acts as a pressurized rod, providing structural support for the embryo and allowing locomotion during larval stages (*Deng et al., 2013*; *Munro et al., 2006*). These early developmental functions of the notochord are conserved from basal chordates to all vertebrates. However, in vertebrates the continuous hollow tube is replaced by a core of vacuolated cells that form a rod within a sheath whose characteristics diverge across vertebrate subgroups. For example, in teleosts such as zebrafish the inner vacuolated cells are surrounded by an epithelial sheath which is encased in a thick extracellular matrix (*Ellis et al., 2013a*; *Ellis et al., 2013b*). On the other hand, in amniotes such as chicken and mice notochord vacuolated cells are surrounded by a seemingly acellular sheath (*Choi et al., 2008*; *Ward et al., 2018*).

Epithelial sheath cells play at least two specialized roles in zebrafish: they provide a template for segmented vertebrae during spine formation (*Lleras Forero et al., 2018*; *Pogoda et al., 2018*; *Wopat et al., 2018*), and they are also able to regenerate vacuolated cells after mechanical damage (*Garcia et al., 2017*; *Lim et al., 2017*; *Lopez-Baez et al., 2018*). While vertebral patterning in the spine occurs via different mechanisms in amniotes compared to teleosts (*Fleming et al., 2015*; *Harris and Arratia, 2018*), the notochord plays a conserved role in attracting osteoblasts that originate in the sclerotome portion of the paraxial mesoderm to form the vertebral bone (*Renn et al., 2013*; *Ward et al., 2018*; *Wopat et al., 2018*). Moreover, vertebral bones develop in a brace-like

fashion around the notochord which acts as a central scaffold in all vertebrates (*Fleming et al., 2015*), suggesting that the notochord plays a conserved structural role in spine formation.

In contrast to sheath cells, notochord vacuolated cells show remarkable similarity in size and structure across diverse vertebrate species such as zebrafish, frogs and pigs (*Garcia et al., 2017*; *Norman et al., 2018*). Each vacuolated cell contains a single fluid-filled vacuole, which is a specialized lysosome-related organelle that occupies most of the cellular volume (*Ellis et al., 2013a*). During zebrafish embryogenesis these vacuoles inflate rapidly and as they expand within the boundaries established by the notochord sheath, they organize in a stereotypical pattern and contribute to the elongation of the embryonic axis (*Ellis et al., 2013b*; *Norman et al., 2018*). Loss or fragmentation of notochord vacuoles causes shortening of the body axis and, later in development, leads to kinking of the spine (*Ellis et al., 2013a*). The axis elongation defect that occurs upon vacuole fragmentation seems to result from a more compact packing of vacuolated cells (*Navis and Bagnat, 2015*); however, it is unclear whether the spine axis defects share the same structural etiology.

In humans, defects causing kinking of the spine axis are relatively common and can occur during gestation, that is congenital scoliosis (CS), or after birth, most commonly during adolescence, that is adolescent idiopathic scoliosis (AIS) (*Boswell and Ciruna, 2017*; *Giampietro et al., 2009*). While CS is widely thought to arise from vertebral segmentation defects (*Giampietro et al., 2009*; *Sparrow et al., 2012*), work in zebrafish has shown that kinking of the notochord sheath due to loss of *col81a1* function results in a CS-like phenotype (*Gray et al., 2014*). In contrast, mutations affecting several different tissues can cause AIS; these tissues include the neural tube (*Grimes et al., 2016*; *Hayes et al., 2014*; *Sternberg et al., 2018*), cartilage (*Karner et al., 2015*), and paraxial mesoderm (*Haller et al., 2018*), as well as potential effects of systemic inflammation (*Liu et al., 2017*). Understanding the cellular mechanisms involved in spine morphogenesis will help elucidate the developmental origin of CS and AIS.

Here, we investigated the role of notochord vacuoles during spine formation in zebrafish, using live imaging, genetic manipulations and forward genetic analyses. Our data show that during spine formation, notochord vacuoles function as a hydrostatic scaffold and normally resist the compressive force generated by concentric vertebral bone growth into the notochord. We found that loss of vacuole integrity, due to genetic manipulation or resulting from loss of *dstyk* function in vacuole membrane fusion, leads to vertebral malformations due to asymmetrical bone growth, resulting in kinking of the spine axis. Thus, we uncovered a role for notochord vacuoles in vertebral patterning and identify a cellular and developmental mechanism that may explain part of the etiology of CS in humans.

## Results

### *spzl* is a recessive mutation that causes notochord vacuole fragmentation, impaired axis elongation and kinking of the spine in zebrafish

Previous work in zebrafish has shown that fragmentation of notochord vacuoles results in kinking of the spine axis during late larval stages (*Ellis et al., 2013a*). However, it is unclear how notochord vacuoles function during spine formation and how this process is affected when vacuoles are fragmented. Mutants that exhibit a robust vacuole fragmentation phenotype in early larvae are affected in essential genes and rarely survive to the spine formation stages (*Ellis et al., 2013a*), limiting the ability to extend these studies into later development. In an unrelated ENU based forward genetic screen, we identified an adult viable recessive mutation that causes both shortening of the embryonic axis and kinking of the spine (*Figure 1*). Because of the short and twisted shape of this mutant, we named it *spaetzle* (*spzl*) after the pasta noodle.

Homozygous *spzl^{s739}* mutants develop a significantly shorter embryonic axis compared to wild-type (WT) siblings (*Figure 1A,B*). Close examination of the notochord in 48 hr post fertilization (hpf) *spzl* mutants by DIC microscopy revealed a pattern consistent with vacuole fragmentation (*Figure 1C*). To confirm this observation, we incubated WT and mutant sibling embryos with the vital dye Cell Trace to visualize intracellular membranes (*Ellis et al., 2013a*). Using live confocal microscopy, we found that by 72 hpf notochord vacuoles were largely fragmented (*Figure 1D*, arrow). To investigate the effect of vacuole defects on notochord organization, we performed a quantitative analysis using 3D renderings (*Figure 1E–H*). We found that the notochord length is significantly

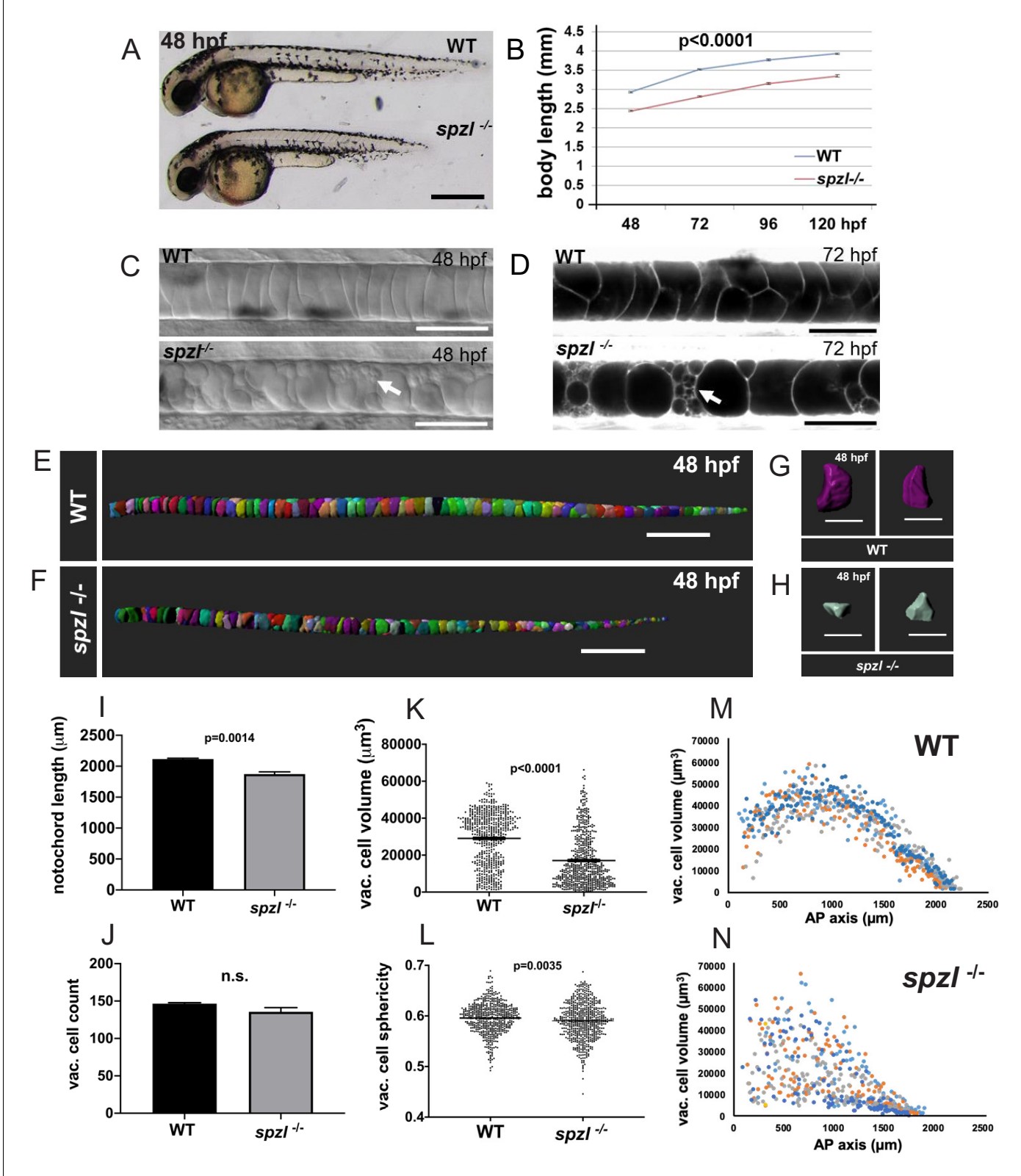

**Figure 1.** *spzl* is a recessive mutation which causes notochord vacuole fragmentation, impaired axis elongation, and altered vacuolated cell packing. (A) Whole mount lateral view of 48 hpf *spzl⁻ᐟ⁻* (bottom) and WT sibling (top) embryos. Scale bar = 500 µm. (B) Body length measurements (mm) from 48 to 120 hpf. n = 30 for WT and n = 27, n = 30, n = 29, n = 28 for *spzl⁻ᐟ⁻* respectively. p<0.0001 at all time points, two-way ANOVA with Sidak's test. At 24 hpf *spzl* mutant embryos (n = 20) are also significantly shorter than WT (n = 15), p=0.001, unpaired t-test using Welch's correction. (C) Live DIC images

*Figure 1 continued on next page*

Figure 1 continued

of 48 hpf WT (top) and *spzl*[-/-] (bottom) embryos. Arrow points to fragmented vacuoles. Scale bars = 50 μm. (D) Live confocal images of 72 hpf WT (top) and *spzl*[-/-] (bottom) notochords stained with Cell Trace to visualize internal membranes. Arrow points to area of vacuole fragmentation. Scale bars = 50 μm. (E–F) Notochord 3D reconstructions for 48 hpf WT (E) and *spzl*[-/-] (F) embryos. Scale bar = 200 μm. (G–H) Single cell 3D reconstructions for WT (G) and *spzl*[-/-] (H) visualized at different angles to show cell shape. Scale bar = 50 μm. (I) Notochord length measurements for WT and *spzl*[-/-] at 48 hpf. (J) Total number of vacuolated cells in WT and *spzl*[-/-] at 48 hpf. (K) Plot of cell volume measurements of WT and *spzl*[-/-] notochord cells at 48 hpf. (L) Sphericity of individual notochord cells for WT and *spzl*[-/-] at 48 hpf. (M–N) Volume of notochord cells along the AP axis of WT and *spzl*[-/-] at 48 hpf. p-values were determined by an un-paired t-test using Welch's correction.

The online version of this article includes the following figure supplement(s) for figure 1:

**Figure supplement 1.** Vacuolated cell arrangement is altered in *spzl* mutants.

shorter in mutants compared to WT (*Figure 1I*). This difference in length was not due to changes in vacuolated cell numbers (*Figure 1J*), but mainly to a significantly smaller vacuolated cell volume in mutants compared to WT (*Figure 1K*). We also observed that cells with fragmented vacuoles typically lose their original ovoid morphology and fill the space in between cells that retain intact vacuoles. To quantify this effect, we first calculated the sphericity index of vacuolated cells, which is the ratio of the surface area of a sphere of equal volume to the surface area of each cell, this ratio gets smaller as the cell shape is distorted from a smooth ovoid shape. We found that sphericity is significantly smaller in mutants compared to WT (*Figure 1L*), indicating a more irregular morphology, also apparent in 3D renderings (*Figure 1G,H*). Then, we plotted the distribution of vacuolated cell volume along the anterior-posterior (AP) and found that in WT it follows a stereotypical pattern whereas in mutants this distribution was randomized (*Figure 1M,N*). These observations are consistent with WT having an ordered arrangement and *spzl* mutants a more compact organization of vacuolated cells. To investigate this possibility we examined the 3D organization of vacuolated cells as reported previously (*Norman et al., 2018*) and found that in *spzl* the stereotypical staircase arrangement of vacuolated cells is largely lost in favor of a more compact and irregular pattern (*Figure 1— figure supplement 1*).

Notably, both the notochord and the AP axis remain straight during embryonic and early larval stages in *spzl* mutants. However, by three weeks post fertilization (wpf), kinking of the AP axis was apparent in most (19 out of 20 animals) *spzl* mutants (*Figure 2A,B*). To determine whether the body axis defects were linked to kinking of the spine we stained fixed 6 wpf larvae with alizarin red to visualize mineralized bone. We found that *spzl* mutants present sharp kinks in their spine axis, in both dorsal-ventral (DV) and left-right axes, typically at two major locations (*Figure 2C,D*, see also *Figure 2—figure supplement 1*). Micro-computed tomography (μCT) of adult fish revealed a similar pattern and allowed detailed 3D visualization of the spine defects (*Figure 2E,F*, see also *Video 1* and *Video 2*). Altogether, these data show that *spzl* mutants develop fragmented notochord vacuoles, have a significantly smaller vacuolated cell volume and show changes in cell organization that lead to a short AP embryonic axis and a kinked spine at late larval stages.

## Loss of Dstyk function causes the *spzl* phenotype

To isolate the *spzl*[s739] mutation, we used whole exome sequencing and positional cloning (*Ryan et al., 2013*). We first mapped the mutation using SNPtrack (*Leshchiner et al., 2012*) to a genomic region on chromosome 22. Then, we circumscribed it by limited positional cloning to an interval of ~1.2 Mb, containing 35 genes including 9 genes closely clustered in 160 kb close to the telomeric end of the critical interval, coinciding with the peak of probability generated by SNPtrack (*Figure 3A,B*). Examination of the sequencing data throughout the critical interval revealed a homozygous nonsense mutation (C1694A) in exon 7 of *dstyk* which is predicted to truncate the encoded protein before the highly conserved kinase domain (*Figure 3C,D* and *Figure 3—figure supplement 1A–B and H*). RT-PCR analysis showed reduced levels of transcript in *spzl*[s739] mutants compared to WT, suggesting the transcript is subject to nonsense mediated decay (*Figure 3—figure supplement 1C–D*). Dstyk is a poorly known dual Ser/Thr and Tyr kinase that has recently been implicated by genetic association studies in renal agenesis in humans (*Lee et al., 2017*; *Sanna-Cherchi et al., 2013*).

Next, we performed whole mount in situ hybridization and found *dstyk* is maternally loaded and expressed in the notochord as early as the 10-somite stage and becomes progressively enriched at

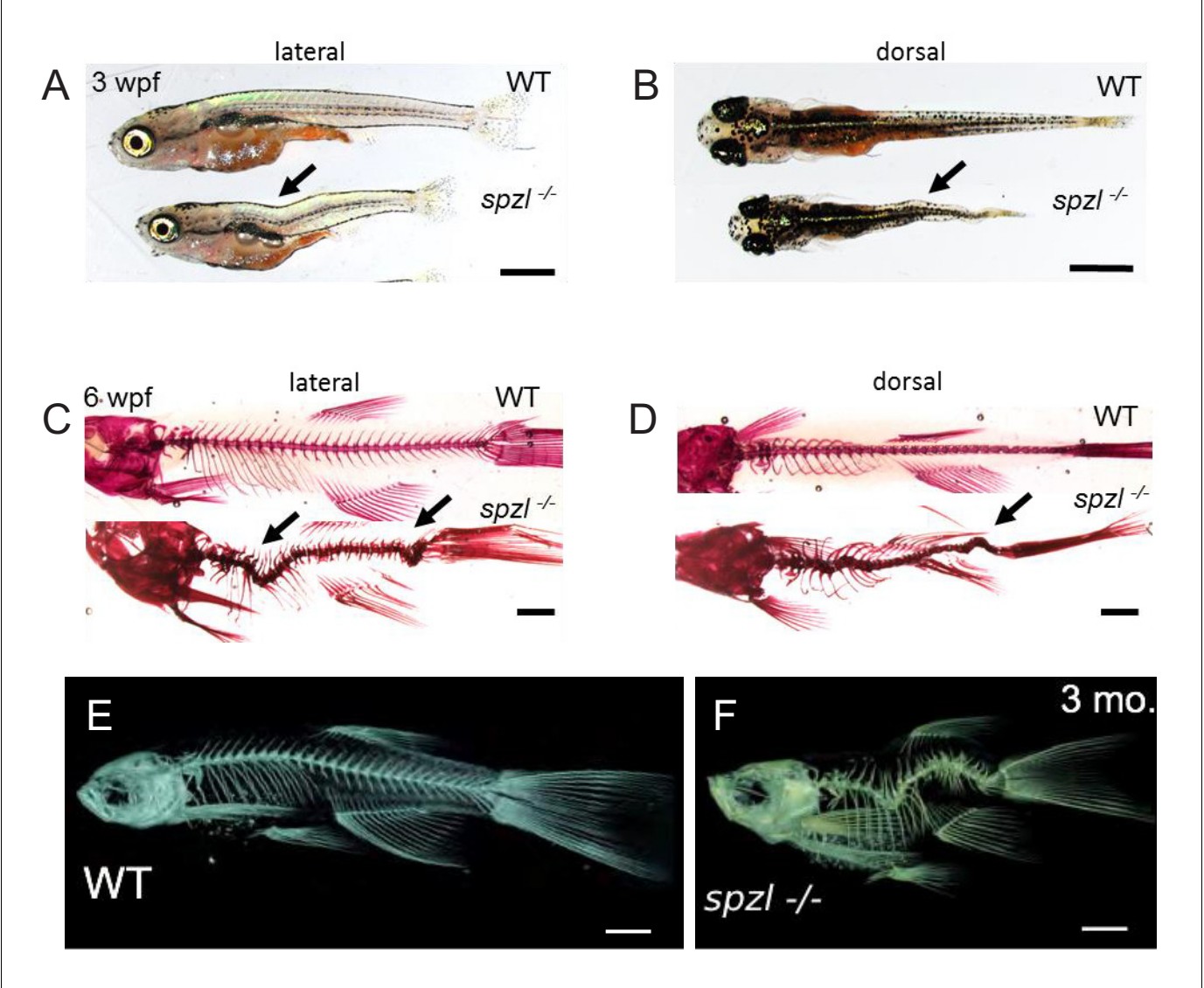

**Figure 2.** *spzl* mutants present spine kinks at juvenile and adult stages. (A–B) Whole mount lateral and dorsal views of 3 wpf WT and *spzl*^-/- fish. Scale bars = 1 mm. (C–D) Lateral and dorsal views of 3 wpf WT and *spzl*^-/- fish stained with alizarin red to visualize the mineralized skeleton. Scale bars = 1 mm. Arrows point to kinks in the axis. (E–F) µCT images of 3 month old WT and *spzl*^-/- adult fish. Scale bars = 2 mm.

The online version of this article includes the following figure supplement(s) for figure 2:

**Figure supplement 1.** Spine defects in *spzl* mutants.

the posterior portion of the notochord by 24 hpf (*Figure 3E*). At that time point the notochord is still developing and newly generated vacuolated cells continue to grow and expand the axis (*Norman et al., 2018*).

To confirm that *dstyk* is indeed the gene mutated in *spzl*^s739, we generated a new allele (*dstyk*^pd1133) using CRISPR/Cas9. We targeted the gRNA to exon 7 and generated a 10 bp deletion that causes a frameshift resulting in an early stop codon in exon 7, before the kinase domain (*Figure 3C*, *Figure 3—figure supplement 1F–G*). Homozygous *dstyk*^pd1133 mutants phenocopy the axis elongation defect of *spzl*^-/-, present fragmented notochord vacuoles as judged by DIC microscopy, and failed to complement *spzl* mutants (*Figure 3F,H*).

Next, we performed rescue assays by injecting a DNA construct expressing WT Dstyk or a kinase-dead version (D757N) in *spzl*^-/- under the control of the *rcn3* promoter, which drives expression in both sheath and vacuolated cells (*Ellis et al., 2013a*). While expression of the WT construct (marked

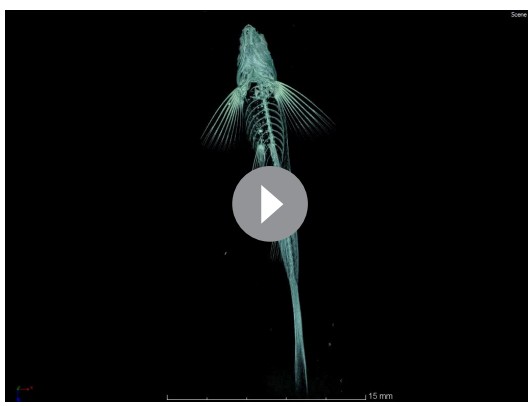

**Video 1.** 3D reconstruction of µCT imaging from a 3 mpf WT fish.
https://elifesciences.org/articles/51221#video1

by GFP co-expression) consistently rescued single vacuole formation (*Figure 4A*), expression of the D757N mutant did not (*Figure 4B*). We also noticed in embryos overexpressing WT Dstyk the presence of enlarged vacuolated cells (*Figure 4A*, left panel) and small cells with vacuoles that might correspond to sheath cells (*Figure 4A*, right panel, arrowhead). To test this possibility, we repeated the experiment in WT embryos expressing a sheath cell marker and found that overexpression of Dstyk leads to vacuole formation in sheath cells (*Figure 4C*), suggesting a role for Dstyk in notochord vacuole biogenesis. To test this idea further, we generated maternal-zygotic *spzl* mutant embryos (*MZspzl*$^{-/-}$) and found a high rate of early embryonic lethality (~96%, n = 330) due to poor egg quality, suggesting a role for Dstyk in oogenesis. Nevertheless, we were able to obtain some

*MZspzl*$^{-/-}$ embryos that survived until notochord formation stages and found that their vacuolated cells were completely devoid of vacuoles, unlike heterozygous and zygotic mutants (*Figure 4—figure supplement 1*). Together, these data show that *dstyk* is the gene mutated in *spzl* and that the kinase activity of Dstyk is required cell-autonomously for notochord vacuole biogenesis and integrity.

## Dstyk regulates fusion of pre-vacuolar carriers with the notochord vacuole

The vacuole fragmentation phenotype of *spzl* mutants is reminiscent of that observed in the HOPS complex mutants *vps11* and *vps18* (*Ellis et al., 2013a*) (*Figure 5—figure supplement 1*). The HOPS complex has been shown to act as a tether for fusion with the vacuole in budding yeast and lysosomes in mammalian cells (*Balderhaar and Ungermann, 2013*). To examine the role of Dstyk in trafficking to the vacuole, we imaged GFP-Rab32a in WT and *spzl*$^{-/-}$ using light sheet microscopy. We focused our attention to the 48–65 hpf time window because it allows clear identification of mutants and vacuole biogenesis is still prominent in the posterior notochord. In the posterior-most part of the WT notochord we observed numerous relatively small vesicles fusing with larger structures that corresponded with the main vacuole (*Figure 5A* and *Video 3*). More anteriorly (i.e. in older vacuoles), we could observe large (~5 µm) pre-vacuolar compartments docking with the vacuole during a long (~2 hr) period in which the docking site was brightly labeled by GFP-Rab32a (*Figure 5C*, arrowheads and *Video 4*), with fusion finally taking place after 2 additional hours (*Figure 5C*, asterisks and *Video 4*). This exceedingly slow fusion process is remarkably similar to that reported recently between vacuolinos (similar to pre-vacuolar compartments referred here) and the tonoplast in plant cells (*Faraco et al., 2017*), and likely reflects the complexity and high energetic cost of these events. In contrast to the dynamics observed in WT, *spzl*$^{-/-}$ mutants exhibited few small vesicles, displayed docked pre-vacuolar compartments (*Figure 5D*, arrowheads and *Video 5*) and no obvious successful fusion events within an 18 hr span (*Figure 5D* and *Video 5*). These data indicate that Dstyk activity regulates fusion of pre-vacuolar compartments with the notochord vacuole.

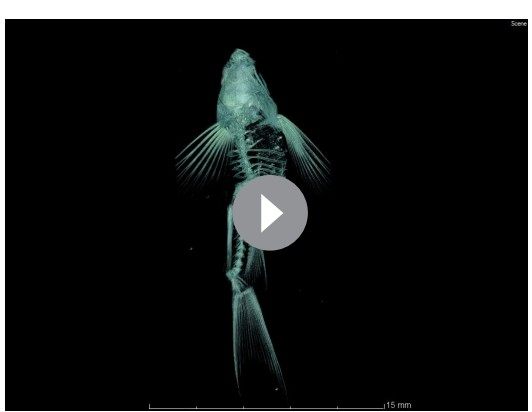

**Video 2.** 3D reconstruction of µCT imaging from a 3 mpf *spzl*$^{-/-}$ fish. Note kinking of the spine axis.
https://elifesciences.org/articles/51221#video2

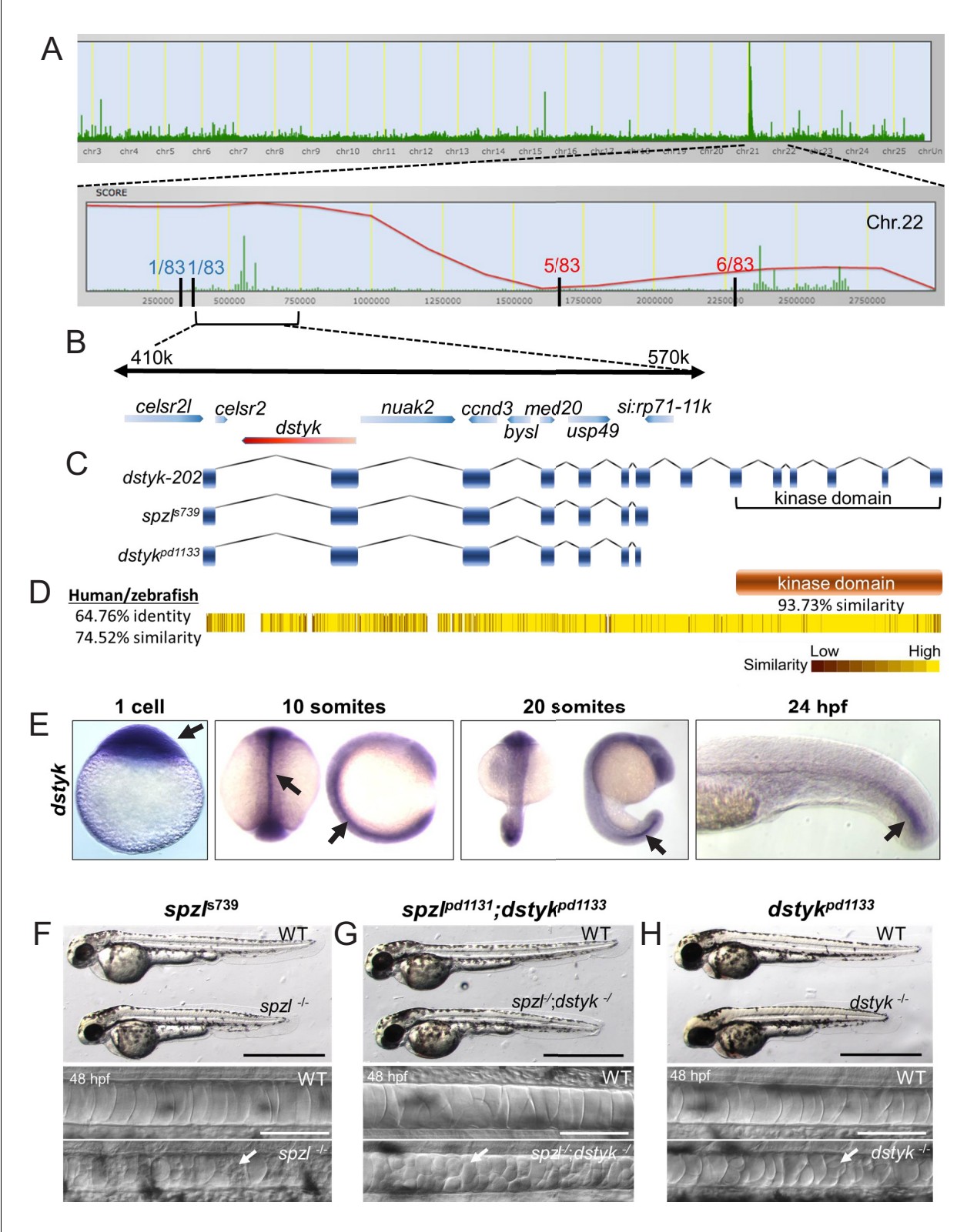

**Figure 3.** Loss of Dstyk function causes the *spzl* phenotype. (**A**) Genome view of SNPtrack linkage analysis of exome sequencing data. The *spzl* locus maps to chromosome 22 (top panel), with a high probability interval (red line) centered around 410–570 Kbps. Recombination events with a panel of 83 mutants at either side of the *spzl* locus define a ~ 1.2 Mb critical interval. (**B**) Genome organization of the critical genomic interval near the *spzl* locus. A nonsense mutation generates an early stop codon in *dstyk*. (**C**) Cartoon depicting predicted gene product structure of *dstyk-202*, *spzl^s739* and the

*Figure 3 continued*
Crispr/Cas9 generated *dstyk^pd1133* allele. (D) Dstyk is a Ser/Thr and Tyr kinase. Heat map depicting amino acid similarity between the human and zebrafish proteins. The kinase domain (highlighted) is 93.73% similar. (E) In situ hybridization for *dstyk* expression at the 1 cell stage, 10 somites, 20 somites, and 24 hpf, respectively. Arrows point to highly expressing areas of the notochord. (F–H) Bright field images of *spzl^-/-* and WT sibling, *spzl; dstyk^pd1133-/-* and WT sibling, *dstyk^pd1133* homozygous mutant and WT sibling, scale bars = 1 mm. DIC images of the notochord of 48 hpf embryos. Arrows point to fragmented vacuoles. Scale bars are 100 µm.
The online version of this article includes the following figure supplement(s) for figure 3:

**Figure supplement 1.** Characterization of *spzl* and *dstyk* mutant alleles.

## Notochord vacuoles absorb compressive vertebral bone growth locally

We showed that in *spzl* mutants, loss of Dstyk kinase activity blocks fusion to the vacuole membrane and that this block is associated with impaired embryonic axis elongation and kinking of the spine axis. Next, to better understand how notochord vacuoles function during spine formation and how their fragmentation is linked to axis kinking, we imaged spine formation live in WT animals. To visualize notochord vacuoles, we used an enhancer trap transgene that expresses GFP specifically in the thin cytoplasm of vacuolated cells (*Yamamoto et al., 2010*), leaving most of the cell volume as a dark space (*Figure 6A*, asterisks). Osteoblasts were visualized using *osx:mcherry-NTR* as a marker (*Singh et al., 2012*). Larvae were staged over the course of ~4 weeks using standard length (*Parichy et al., 2009*) as a reference.

Initially (4.5 mm stage), the contours of the notochord were straight and the vacuoles appeared fully inflated and of ovoid shape, remaining roughly unchanged until 6.2 mm. Then, starting around 6.9 mm we observed the initial manifestations of notochord indentations (*Figure 6A*, arrowheads) and the re-orientation of vacuoles (arrows) in a direction parallel to the AP axis of the spine. This process became more obvious as bone formation proceeded, with vertebral bone (centra) reaching a characteristic hourglass shape by 8.7 mm (*Figure 6A*, right panels). At this point, vacuoles underneath the central portion of each centrum appeared stacked and flat; whereas, vacuoles found underneath the ends of centra were fragmented, tightly packed and bulged into the prospective intervertebral space (*Figure 6A*, bottom panels). These data show that during spine morphogenesis as more osteoblasts are added and mineralization increases, vertebral bones grow concentrically, compressing the notochord. During this process the centra acquire an hourglass shape and the vacuolated cells seem to react locally to the compressive force generated by bone ingrowth by re-organizing into two distinct arrangements (*Figure 6B,D*). To track changes in vacuolated cell shape quantitatively we calculated the sphericity index of vacuolated cells over the spine formation window using 3D renderings and found that it presents a sigmoidal behavior with a plateau during chorda-centra and initial bone formation stages of (9–12 dpf), which then drops again as vacuolated cells underneath centra are flattened and become stacked during bone growth (*Figure 6E*). These direct observations in live zebrafish are consistent with histological snapshots from spine morphogenesis in mice (*Choi and Harfe, 2011*) and salmon (*Kryvi et al., 2017*; *Wang et al., 2013*), underscoring the conservation of the process.

Next, we investigated how notochord vacuole fragmentation affects vertebral bone formation. To this end we expressed a dominant-negative form of Rab32a (DN-GFP-Rab32a) in vacuolated cells to fragment vacuoles. We chose this strategy over vacuolated cell ablation because following ablation vacuolated cells are regenerated by trans-differentiation of sheath cells (*Garcia et al., 2017*; *Lopez-Baez et al., 2018*). Upon expression of DN-GFP-Rab32a in one small cluster of vacuolated cells we observed, as shown before (*Ellis et al., 2013a*), that vacuoles fragmented in those cells. This was associated with increased indentation and bending of the notochord in those areas as centra growth proceeded (*Figure 6F*, arrows). Interestingly, at those locations we also observed that centra were malformed (*Figure 6F*, dotted lines) and of irregular size compared to nearby centra that did not overlay fragmented vacuoles.

Together, these data show that centra growth compresses the notochord and that notochord vacuoles normally absorb locally the compressive force generated by vertebral bone growth.

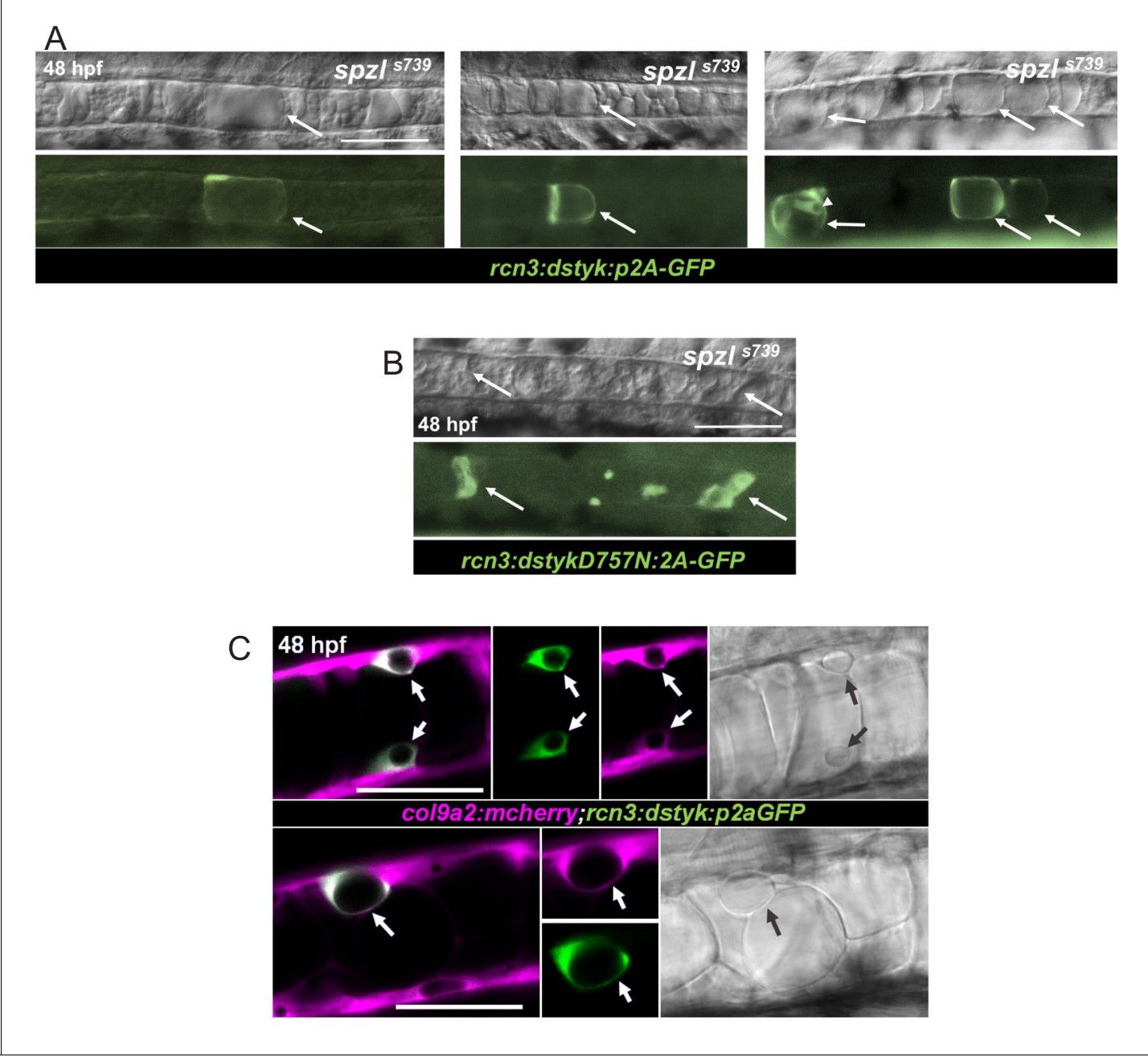

**Figure 4.** Dstyk's kinase activity is required cell-autonomously for vacuole formation. (**A**) Live DIC image (top) and live fluorescent image (bottom) of *spzl* mutants injected with a construct driving *dstyk:p2aGFP* expression in the notochord. The arrows point to GFP expressing cells with rescued vacuole integrity. Arrow head points to an expressing sheath cell that also develops a vacuole. WT *dstyk* rescued vacuole integrity in 90% of expressing cells (n = 30). Scale bar = 100 μm. (**B**) Live DIC image (top) and fluorescent image (bottom) of a *spzl⁻ᐟ⁻* embryo injected with *dstykD757N:p2aGFP*, a construct driving a (D757N) kinase dead version of *dstyk* in the notochord. Expressing cells (arrows) exhibit fragmented vacuoles and only 5% rescue (n = 20). Scale bar = 100 μm. (**C**) Confocal and DIC images of WT embryos at 48 hpf expressing *col9a2:mcherry* in the notochord sheath and injected with *dstyk:p2aGFP*. Sheath cells expressing *dstyk:p2aGFP* develop a vacuole. Arrows point to the vacuole. Scale bar = 50 μm.
The online version of this article includes the following figure supplement(s) for figure 4:

**Figure supplement 1.** Maternal zygotic embryos completely lack inflated vacuoles.

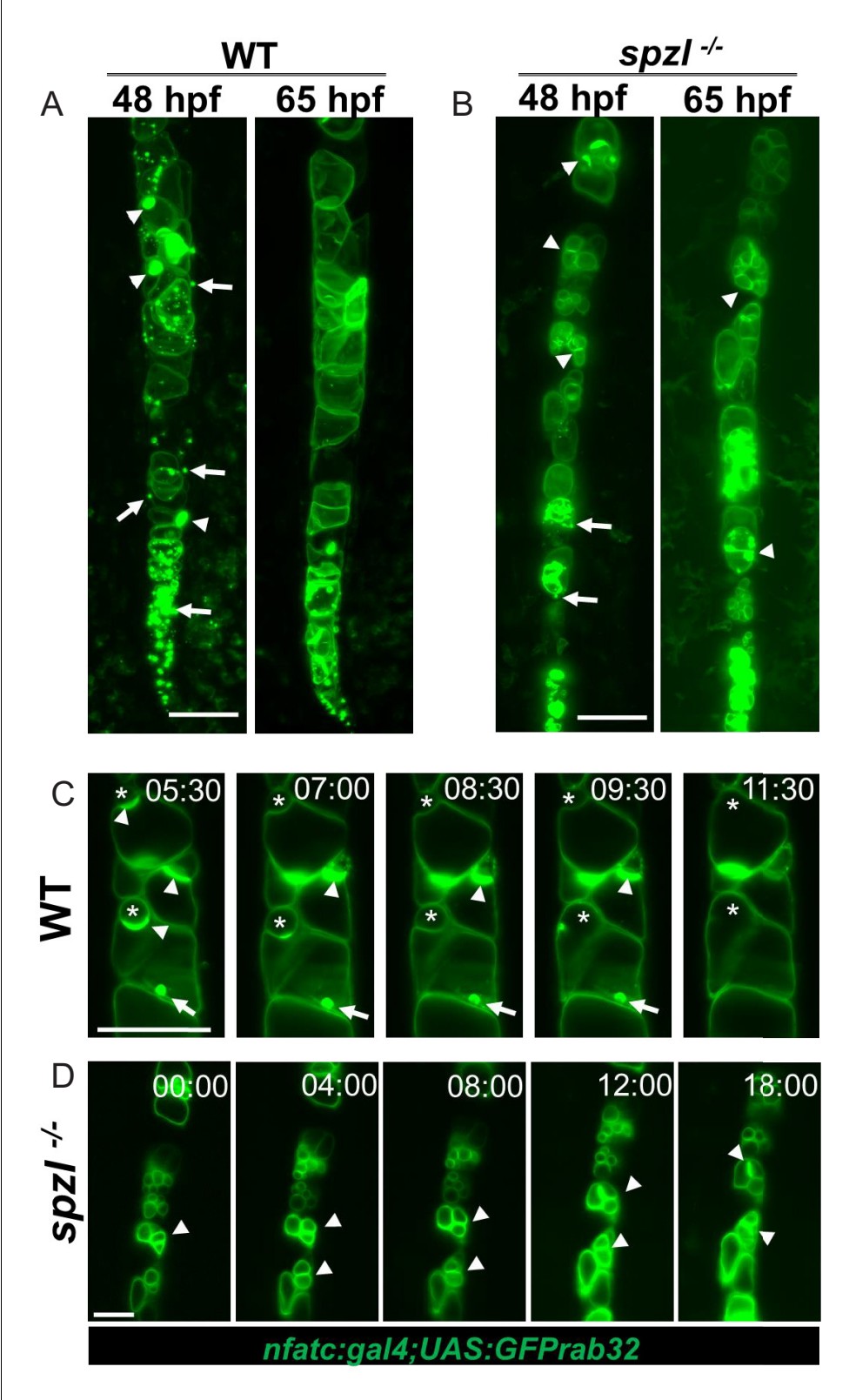

**Figure 5.** Dstyk regulates fusion of pre-vacuolar carriers with the notochord vacuole. (A–B) Maximum projection of a light sheet microscopy live image of WT and *spzl*⁻/⁻ embryos at 48hpf and 65hpf, respectively. (C–D) Still images from a 17 hr light sheet movie of a WT and *spzl*⁻/⁻ embryos zebrafish embryo during vacuole biogenesis depicting vacuole fusion events. Vacuole membranes are labeled by *NFATC:gal4;UAS:GFPrab32* expression. Arrows

*Figure 5 continued*

point to GFP-Rab32 positive vesicles. Arrowheads point to large vesicle (vacuolinos) docking events. n = 2 animals per genotype. Asterisks mark fusion events. Scale bar = 100 µm.

The online version of this article includes the following figure supplement(s) for figure 5:

**Figure supplement 1.** *vps18* mutants exhibit a vacuole fragmentation phenotype similar to that of *spzl* mutants.

## Loss of notochord vacuole integrity reduces notochord rod pressure and leads to spine kinking in *spzl* mutants during vertebral bone growth

We have shown that in *spzl* mutants notochord vacuoles exhibit membrane fusion defects and fragment. We also showed that vacuole fragmentation leads to kinking during spine formation. To visualize this process over time we imaged spine formation in WT and *spzl* mutants using the same markers for vacuolated cells and osteoblasts as shown in *Figure 6*. We observed that initially numerous vacuoles were still present, likely due to maternal load of *dstyk,* and the notochord remained straight in *spzl* mutants (*Figure 7A,B*, top panels). Then, vacuoles became progressively smaller and eventually disappeared, as judged by the labeling of the whole cell with cytoplasmic GFP. However, the notochord and developing spine remained straight until bone deposition progressed further, at which point sharp clefts could be observed in the notochord and spine kinks developed associated with malformed centra (*Figure 7B*, bottom panel, arrow). We were able to observe snapshots of this process in a single mutant animal (*Figure 7C–E*) in which areas of the notochord that retained vacuoles had a straight spine (*Figure 7F*); whereas, in other areas where vacuoles had been lost the notochord was highly squeezed or absent and the spine was sharply kinked (*Figure 7G–I*). This coexistence of seemingly different developmental stages and degrees of phenotypic manifestation likely reflects the temporal development of the spine in the AP direction, with older more anterior vacuolated cells likely retaining more maternal *dstyk* than those at the more caudal regions of the axis. Importantly, vertebral malformations in *spzl* mutants were not due to segmentation defects, since notochord sheath segmentation preceding vertebral bone formation (*Wopat et al., 2018*) occurred normally as in WT (*Figure 7—figure supplement 1*).

The effect of experimental vacuole fragmentation on spine formation (*Figure 6F*) and the tight correlation between loss of notochord vacuoles, centra morphology and spine kinking suggested that the notochord of *spzl* mutants loses its internal pressure as notochord vacuoles fragment and are lost, rendering the notochord less stiff and unable to resist compression. Because measuring notochord stiffness directly in vivo is currently unfeasible and devices such as pressure gauges would not be useful for the notochord as the core is not a continuous lumen, we looked instead for ways to estimate the relative internal pressure of the notochord. Comparing live and histology images of the notochord we noticed that in WT the nuclei become compressed into bent disks as vacuoles inflate, whereas in *spzl* mutants nuclei remain spheroid in shape (*Figure 8A,B*). We reasoned that as the notochord sheath forms a fairly rigid corset, inflation of vacuoles enlarges the cells, eventually displacing the nuclei and compressing them into bent disks. We therefore plotted the distribution of vacuolated cell nuclei within the notochord in histological sections of 14 dpf larvae, just prior to the onset of centra formation, and found that while in WT the nuclei are highly clustered at the periphery, in *spzl* mutants their localization is

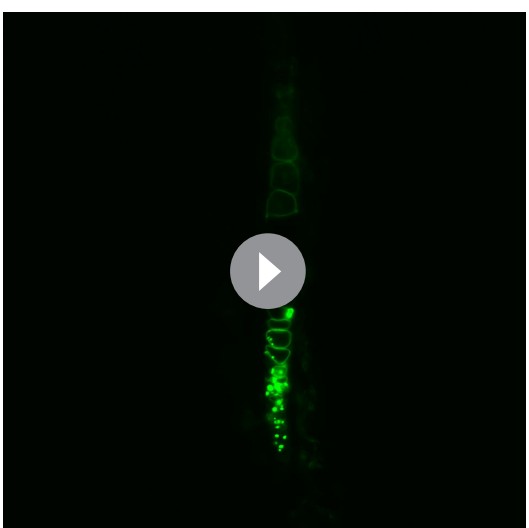

**Video 3.** Live imaging of vacuole biogenesis in WT. Light sheet imaging of GFP-Rab32a membranes in a 48 hpf WT embryo. Note small carriers fusing with the main vacuole.

https://elifesciences.org/articles/51221#video3

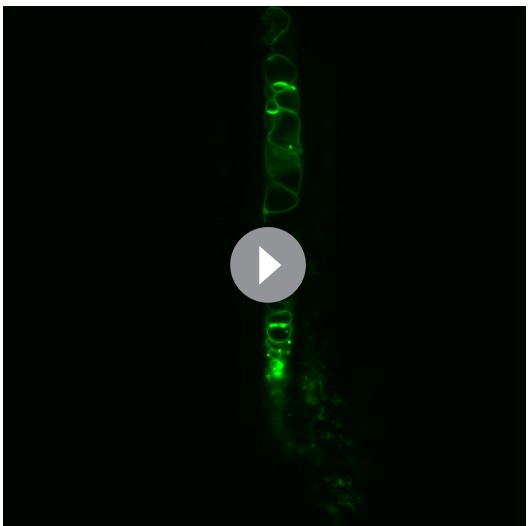

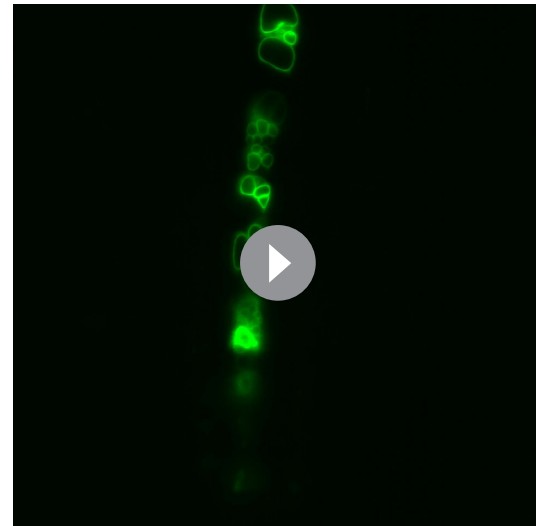

**Video 4.** Slow fusion of prevacuolar membranes with notochord vacuoles. Light sheet imaging of GFP-Rab32a membranes in a 48 hpf WT embryo. Large pre-vacuolar membranes, i. e. vacuolinos, fuse with the main vacuole after a long lag period.
https://elifesciences.org/articles/51221#video4

**Video 5.** Live imaging of vacuole biogenesis in spzl$^{-/-}$. Light sheet imaging of GFP-Rab32a membranes in a 48 hpf spzl$^{-/-}$ embryo. Fusion of small and large carriers to the main vacuole is impaired.
https://elifesciences.org/articles/51221#video5

randomized (*Figure 8C*). Moreover, while nuclear volume remains constant (*Figure 8D*), nuclear shape is significantly more spherical in *spzl* mutants compared to WT (*Figure 8E–G*). These data indicate that fragmentation and loss of notochord vacuoles in *spzl* mutants leads to reduced internal notochord pressure, likely resulting in a less stiff and more easily deformable structure.

Our experimental manipulations and live imaging of *spzl* mutants suggested that upon loss of vacuoles the spine kinks as a result of vertebral bone growth. To demonstrate this is indeed the case we first imaged single WT and *spzl* mutants over the spine formation process. We observed that, as indicated above, the notochord was initially straight (*Figure 9A,B*, top panels). However, as bone formation proceeded, visualized by live calcein staining, we observed the development of a kink in the spine that became progressively shaper over time (*Figure 9A,B*). As *spzl* mutants typically develop a major kink near the posterior end of the swim bladder (*Figure 2—figure supplement 1*) we could follow different animals and quantify the progression of those kinks. We found that in individual mutants we could follow through the whole process, the spine kinks became increasingly sharper with the same relative progression (*Figure 9C*) along with vertebral bone growth.

Next, we wanted to determine how the effect of vertebral bone growth impacts the notochord and how the response of the notochord to compression relates to centra and spine axis formation. To this end we first measured the shape of the notochord at ~28 dpf along the AP axis in WT and *spzl* mutants using a simple symmetry index which we calculated by fitting a circle on the dorsal and ventral curvature of notochord underneath each centrum (*Figure 9D,E*). This index approaches −1 if the notochord is compressed symmetrically. While in WT the notochord was compressed symmetrically throughout the AP axis, *spzl* mutants showed wide asymmetries that changed in orientation respect to the DV axis at two major points (*Figure 9F*). Interestingly, those areas in which the direction of notochord asymmetry changed in *spzl* mutants were also associated with the most frequent position of sharp kinks in the spine (*Figure 9F*, arrows).

## Centra mineralization and notochord morphology are tightly linked

To more accurately determine how notochord vacuole integrity defects impact vertebral bone formation, we next performed a high resolution μCT analysis. In 3D reconstructions we observed asymmetrically shaped central canals in the centra corresponding to kinked areas of the spine in *spzl* mutants (*Figure 10A,B*, dotted areas). In single imaging planes, we could clearly observe that the centra in those areas were also asymmetrically shaped in *spzl* mutants compared to WT

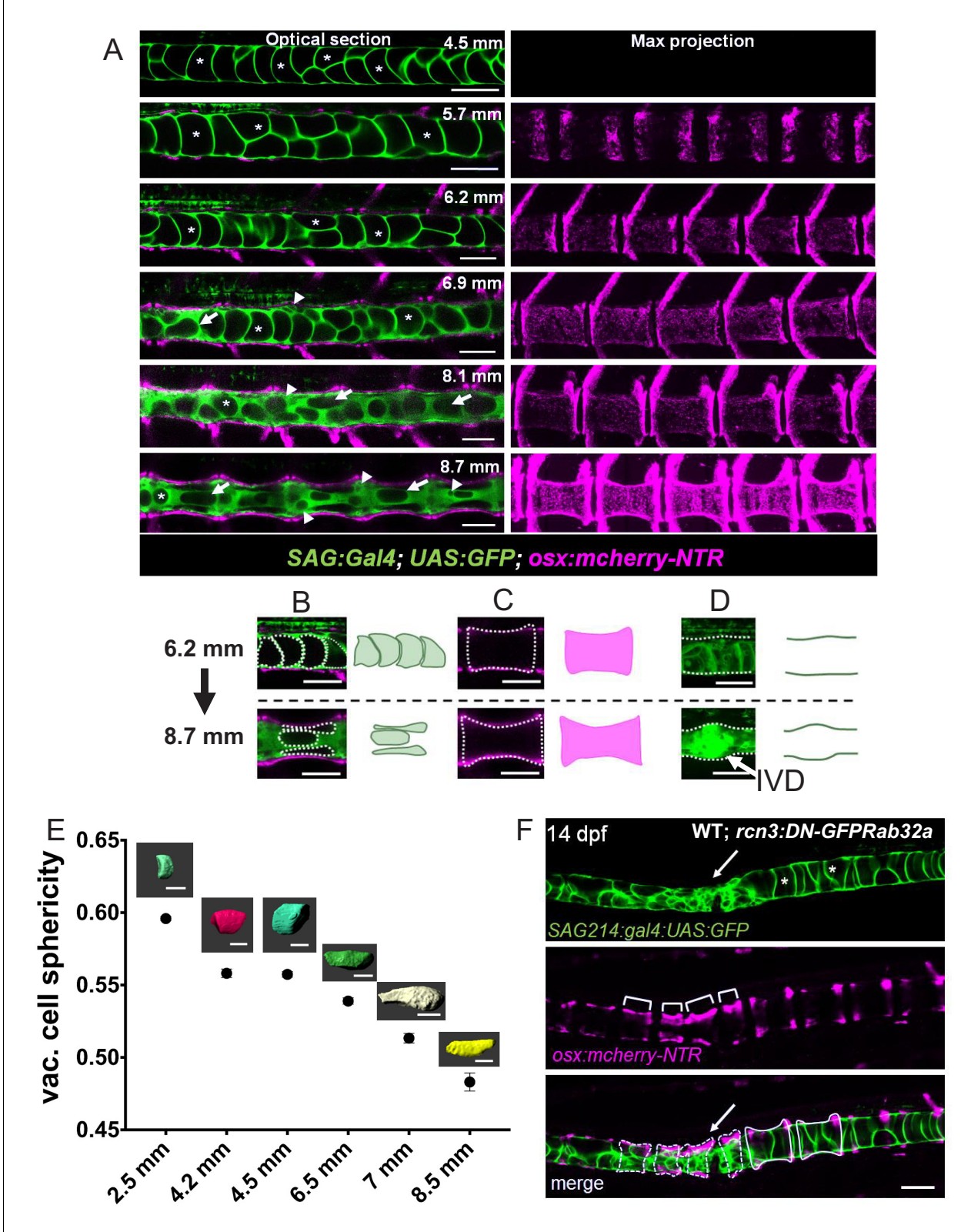

**Figure 6.** Notochord vacuoles absorb compressive vertebral bone growth. (A) Optical sections (left panels) of a live confocal time course during spine formation in WT fish expressing GFP in the vacuolated cells of the notochord and mCherry-NTR in osteoblasts. Asterisks mark the lumen of large vacuoles. Arrows point to vacuoles that re-orientate as they are being squeezed by the growing bone. Maximum intensity projection (right panel) of osteoblast recruitment during bone development. The indentations in the notochord result from concentric bone growth. n = 5, 8, 3, 3, 3, 2 animals

*Figure 6 continued on next page*

*Figure 6 continued*

respectively. (**B–D**) Schematic depicting the shape changes that occur in vacuoles (**B**), centra (**C**), and the IVD (**D**) during vertebral growth between the standard-length stages 6.2 mm and 8.7 mm. Arrowheads point to areas of the notochord with fragmented vacuoles. Arrows indicate vacuoles undergoing shape change. Asterisks mark intact vacuoles. (**E**) Average sphericity of vacuolated cells decreases during notochord development and spine formation stages. Examples of cellular shapes are illustrated above each time point on the graph. Scale bar = 50 μm. (**F**) Live confocal image of a 14 dpf larva expressing *DN-GFP-Rab32a* mosaically in the notochord. n = 4 animals. Brackets mark centra. Dotted lines outline malformed centra. Solid lines outline normal centra. *Tg(SAGFF214A:gal4); Tg(UAS:GFP); Tg(osx:NTR-mCherry)*. Scale bars are 100 μm.

(*Figure 10C–E*). Interestingly, when we compared relative bone density throughout the skeleton we found that it is selectively increased in the spine of *spzl* mutant compared to WT (*Figure 10F–I*), suggesting that bone mineralization is regulated by the notochord. This could be due to a role for *dstyk* in osteoblasts, but we could not detect the transcript in osteoblast around the notochord (*Figure 10—figure supplement 1*).

The link between loss of notochord vacuoles and increased centra mineralization prompted us to investigate whether the converse is also true, that is that increased mineralization can induce notochord vacuole loss. To this end we treated WT larvae with retinoic acid (RA) to induce premature and increased centra mineralization as shown previously (*Laue et al., 2008*; *Spoorendonk et al., 2008*). Treatment of larvae for one week, starting at 14 dpf, with 100 nM RA led to increased centra mineralization as expected and also caused premature stacking of vacuolated cells in 6 mm larvae compared to DMSO controls (*Figure 11A,B*). Later on, as mineralization progressed even further notochord vacuoles were almost completely lost and the notochord was highly compressed in RA treated compared to DMSO controls (*Figure 11C,D*).

To explore the relationship between centra mineralization and notochord morphology further we examined *entpd5* (*nob*) mutants, which are unable to mineralize (*Huitema et al., 2012*). To this end we followed notochord vacuole morphology and bone mineralization using Cell Trace and alizarin red staining respectively in live WT and *nob* mutants and found that loss of mineralization reduces notochord deformation and prevents vacuolated cell stacking (*Figure 11E,F*). At the 9.5 mm stage when mineralization of centra was advanced in WT, but absent in mutants, using histological sections we could readily detect vacuolated cell rearrangement into centra and IVD areas in WT, but not in *nob* mutants. These data indicate that bone mineralization leads to compression of the notochord and vacuole rearrangement.

Together, these data show that upon fragmentation or loss of notochord vacuoles, the notochord presents a phenotype consistent with reduced internal pressure. This renders the notochord more easily deformable, leading to asymmetric compression of the notochord during vertebral bone growth, ultimately causing vertebral malformation and spine kinking. Altogether, our findings reveal a crucial mechanical role for notochord vacuoles in proper spine formation during vertebral bone growth.

## Discussion

In this study we have investigated the role of notochord vacuoles during spine morphogenesis. Using the experimental advantages of the zebrafish model, we showed that the fluid filled vacuoles allow the notochord to function as a hydrostatic scaffold during spine morphogenesis. Our studies revealed that as the vertebral bones grow concentrically into the notochord, vacuoles absorb compressive centra growth symmetrically and locally by rearranging and fragmenting (*Figures 6* and *11*). When notochord vacuoles are fragmented or lost prior to bone formation, the notochord rod appears to lose its internal pressure and deforms asymmetrically during centra growth, leading to vertebral malformations and kinking of the spine axis (*Figures 6*, *8* and *9*). These results uncover a role for notochord vacuoles in vertebral bone morphogenesis and suggest CS in humans can develop independently of vertebral segmentation defects.

The stereotypical arrangement of vacuolated cells within the confinement of the tubular notochord sheath (*Norman et al., 2018*), embryonic axis elongation (*Ellis et al., 2013a*), and symmetric vertebral bone growth during spine formation (this work) all critically depend on the mechanical properties of the notochord, which are largely imparted by the behavior of its vacuoles. When fusion of membranes to the vacuole is impaired during embryogenesis (e.g. *spzl* mutants), vacuolated cells

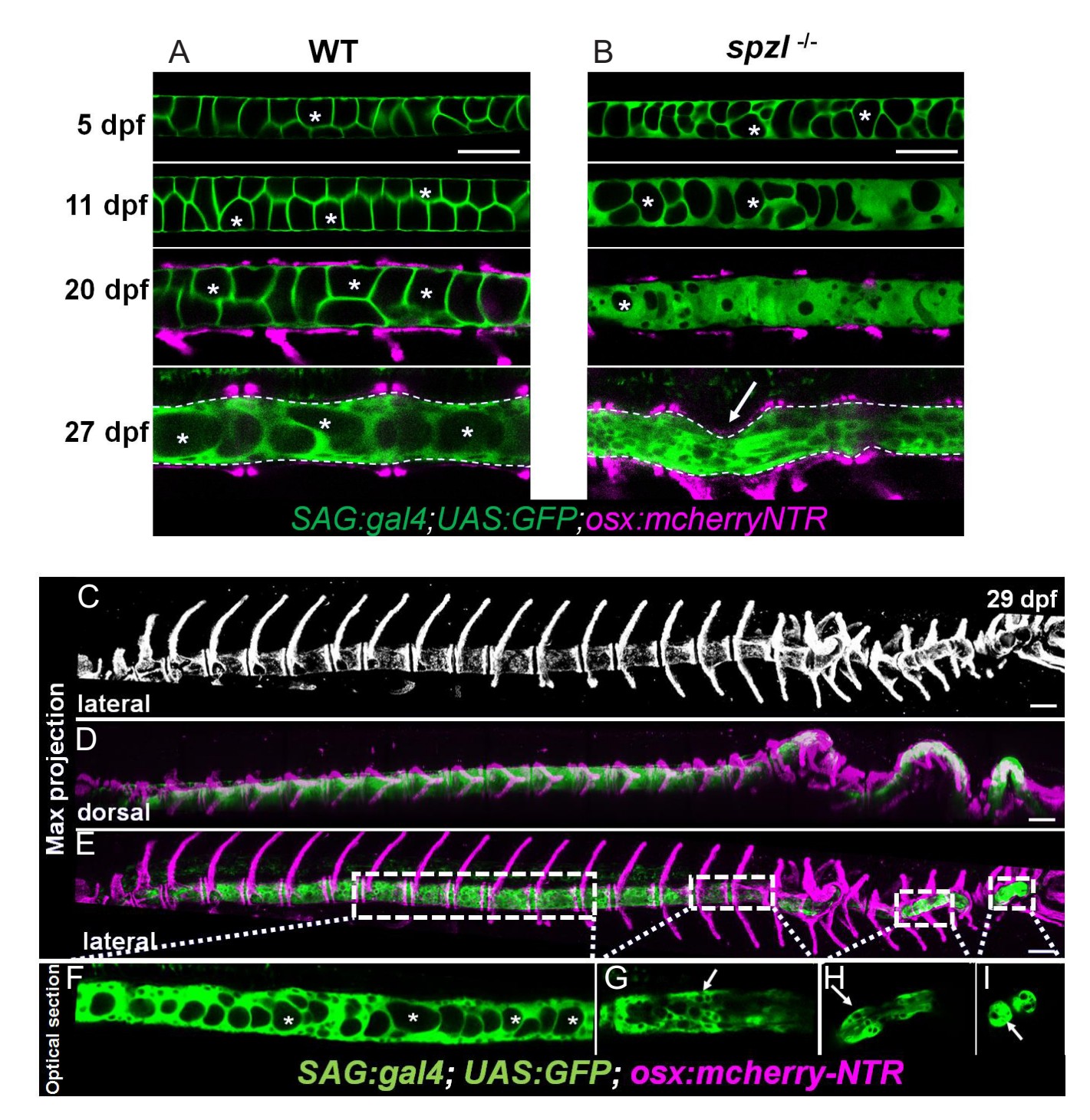

**Figure 7.** Loss of notochord vacuole integrity is associated with spine kinking in *spzl* mutants. (**A**) Live confocal imaging time course of vacuole dynamics and morphology during spine development in WT zebrafish larvae. n = 3, 6, 11, 13. (**B**) Live confocal imaging time course of vacuole dynamics and morphology during spine development in *spzl*[-/-] larvae. n = 3, 4, 8, 7. (**A–B**) Larvae express GFP in the cytoplasm of vacuolated cells and mCherry in osteoblasts. Note the progressive loss of vacuoles in *spzl*[-/-] and the straight notochord and spine prior to the onset of concentric vertebral growth. Dotted lines mark the edge of the notochord. Arrow points to a spine kink in *spzl*[-/-]. (**C**) Maximum intensity projection of a laterally oriented 29 dpf *spzl*[-/-] larva. The spine is visualized by mCherry expression in the osteoblasts. (**D–E**) Maximum intensity projections showing dorsal (**D**) and lateral (**E**) views of a 29 dpf *spzl*[-/-] larva expressing mCherry in the osteoblasts and GFP in the notochord. The anterior spine is straight while the posterior region has developed several spine kinks (**F**) Optical confocal section of the straight anterior portion of the spine. Asterisks mark intact vacuoles. (**G–I**) Optical

*Figure 7 continued on next page*

*Figure 7 continued*

confocal sections of the posterior notochord where vacuoles are highly fragmented or absent and the spine is highly kinked. *Gt(SAGFF214A:gal4); Tg (UAS:GFP); Tg(osx:NTR-mCherry)*. Scale bars are 100 μm.

The online version of this article includes the following figure supplement(s) for figure 7:

**Figure supplement 1.** Notochord segmentation occurs normally in *spzl* mutants.

do not fully expand and vacuoles fragment further, allowing cells to re-arrange more easily within the notochord (*Figure 1* and *Figure 1—figure supplement 1*). This results in more tightly packed cells within a rod that seems to have low internal pressure and thus a limited ability to support axis elongation and proper spine formation (*Figures 1*, *8* and *9*). In our working model of how notochord vacuoles functions during vertebral bone growth (*Figure 12*), we propose that the capacity to fragment during spine formation is key for cushioning vertebral bone ingrowth locally, thereby allowing symmetric centra formation. Conversely, early loss or fragmentation of vacuoles renders the notochord less stiff and more prone to deformation and unable to absorb bone growth locally during spine formation.

The occurrence of kinks at two major sites along the AP axis of the spine when vacuoles are lost globally (*Figure 9F* and *Figure 2—figure supplement 1*) indicates that the entire structure becomes mechanically coupled into one hydrostatic mechanism in *spzl* mutants. This behavior may help explain the change in notochord design from the ancestral continuous hollow tube found in early chordates to the vertebrate structure in which the lumen is compartmentalized into multiple vacuoles. In addition to the local mechanical effects, it is also important to note that the exact location, orientation and magnitude of deformations occurring in the notochord and spine are likely also influenced by organism-wide mechanics. These include locomotion and the spatial constraint of other organs such as the swim bladder that can alter the way forces act on the axis. It will be interesting to pursue these ideas by building physical and theoretical models that may allow more precise definition and understating of the mechanical forces acting on the notochord during spine formation.

Our results clearly show that vacuole fusion regulated by Dstyk kinase activity is a critical cellular process that impacts whole axial development. One possibility is that Dstyk controls fusion of membrane with the vacuole by regulating a direct effector of fusion such as Vps41, which in yeast regulates fusion with the vacuole membrane (*Balderhaar and Ungermann, 2013*). However, the phosphorylation site in Vps41 crucial for fusion is not conserved in zebrafish and we have yet not been able to identify alternative candidate effectors regulated by Dstyk. Alternatively, it is also possible that Dstyk acts more indirectly as a master regulator via transcriptional control. Nevertheless, our imaging data indicate that the last step of fusion with the vacuole is highly complex, taking up to 4 hr (*Figure 5*), and likely represents the limiting step in vacuole biogenesis and maintenance. On the other hand, the differential behavior of vacuolated cells beneath growing centra and in the prospective intervertebral space (*Figures 6* and *11*) also suggests that vacuole fragmentation is selectively regulated during spine formation.

One important conclusion of our study is that vertebral patterning is not solely dependent on vertebral segmentation, which occurs normally in *dstyk* mutants (*Figure 7—figure supplement 1*). Thus, vertebral malformations, and likely also CS in humans, can result from altered paraxial mesoderm segmentation (*Sparrow et al., 2012*), defective mechanical regulation (this study), or alterations in the notochord sheath that produce a kinked scaffold without directly altering vertebral segmentation (*Gray et al., 2014*). These findings suggest that while the notochord structure and specific processes such as segmentation may differ across different vertebrate groups (*Fleming et al., 2015*), core mechanical roles of the notochord in vertebral and spine axis patterning are conserved. Our results also suggest that the mineralizing activity of osteoblasts is subject to mechanical feedback and that notochord morphology and centra mineralization are crossregulated (*Figures 10* and *11*). Understanding how this process is controlled may ultimately allow more effective correction of spine defects.

Previous human genetic studies have linked *DSTYK* to renal agenesis (*Lee et al., 2017*; *Sanna-Cherchi et al., 2013*). However, we did not detect overt kidney defects in zebrafish *dstyk* mutants (*Figure 9—figure supplement 1*). This discrepancy may be due to the fact that the variants reported

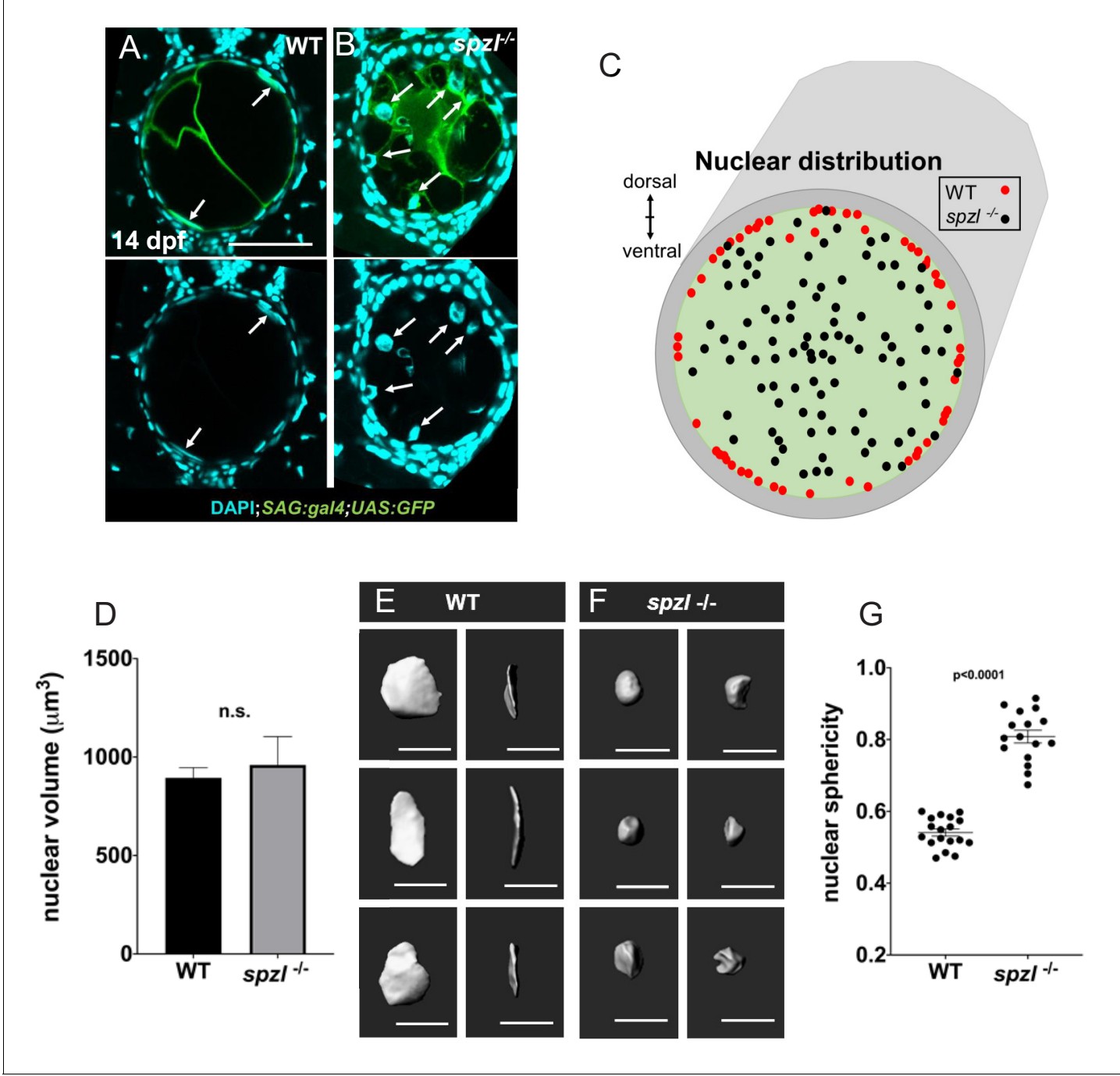

**Figure 8.** Nuclear distribution and shape indicate that hydrostatic pressure of the notochord is decreased in *spzl* mutants. (**A–B**) Confocal images of cross sections of 14 dpf WT and *spzl*$^{-/-}$ larvae section. Vacuolated cells are labeled with *SAG:gal4;UAS:GFP* and nuclei are stained with DAPI. Scale bar = 50 μm (**C**) Plot of vacuolated cell nuclear distribution for WT (red, n = 29) and *spzl*$^{-/-}$ (black, n = 32). (**D**) Nuclear volume in WT and *spzl*$^{-/-}$ at 14 dpf. (**E–F**) Reconstructions of nuclei in WT (**E**) and *spzl*$^{-/-}$ (**F**) and two different viewing angles. Scale bar = 20 μm (**G**) Sphericity of WT and *spzl*$^{-/-}$ nuclei at 14 dpf. p-values were determined by an un-paired t-test using Welch's correction.

in humans do not affect the kinase domain and could thus reflect other developmental roles of this protein. It is also important to consider that genetic association studies cannot definitively assign causality to genetic variants and there is some controversy about the pathogenicity of *DSTYK* variants (*Heidet et al., 2017*). Moreover, genetic association studies may also be compounded by complex genetic interactions with other genes that may be different or not present in our experimental

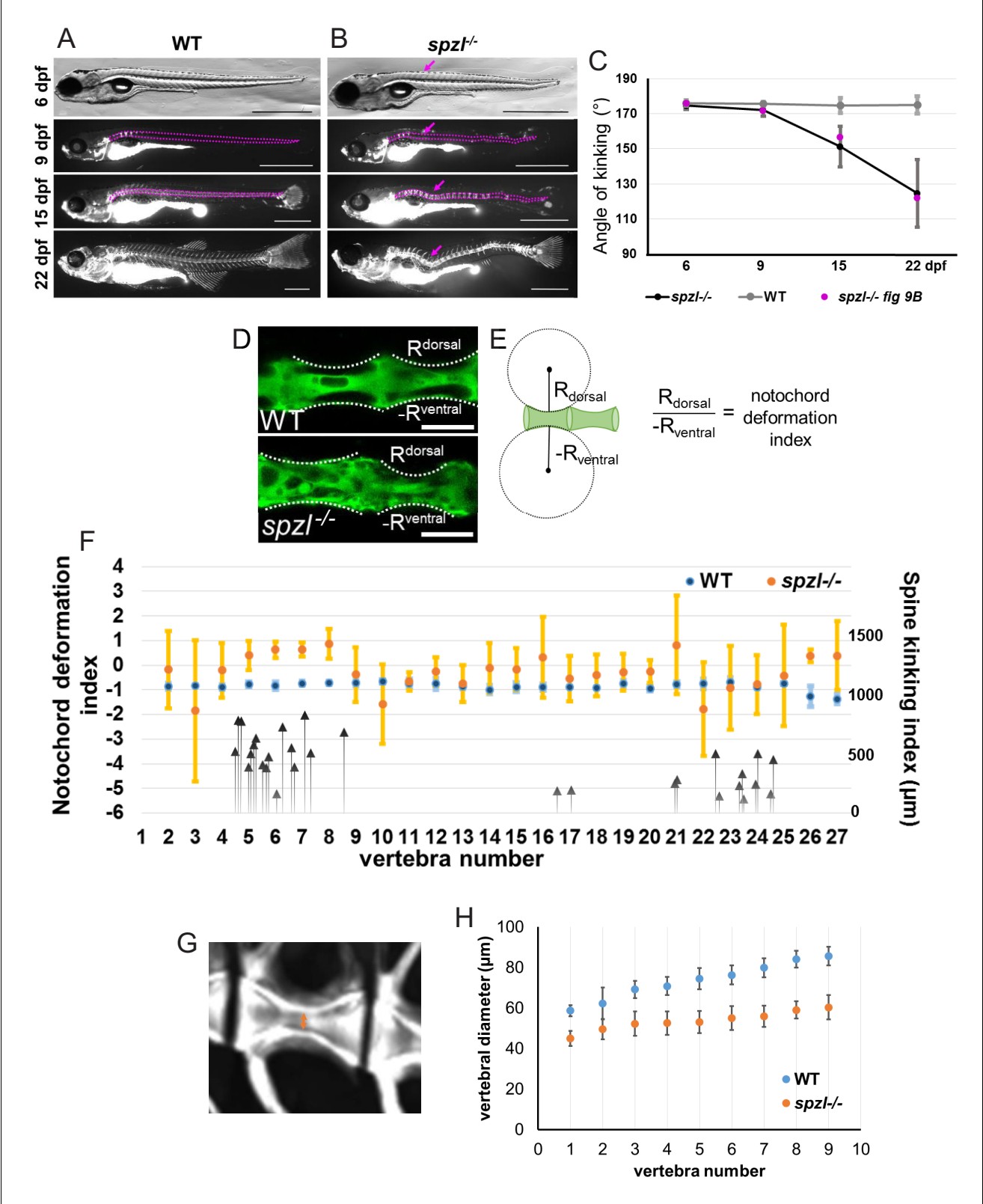

**Figure 9.** Loss of notochord vacuole integrity leads to spine kinking in *spzl* mutants during vertebral bone growth. (A–B) Individual (A) WT and (B) *spzl*$^{-/-}$ fish were incubated with calcein and imaged under a fluorescence compound microscope throughout the time course of spine development. The dotted lines mark the notochord and spine and the magenta arrows mark developing kinks in the spine over time. The solid magenta lines mark that area where we measured the angle of kinking. Scale bars are 1 mm. (C) Quantification of the angle of spine kinking over time. Magenta dots refer to

*Figure 9 continued on next page*

*Figure 9 continued*

the representative fish in panel (**B**). n = 3 for WT, n = 18 for *spzl*^-/-. Note that axis kinking starts after the onset of mineralization and gets progressively sharper as vertebral bone grows. (**D**) Optical confocal section of WT (top) and *spzl*^-/- (bottom) expressing GFP in the cytoplasm of notochord vacuolated cells. Dotted line label the deformation angle caused by the growing vertebrae. The radius of this angle was then used to determine the notochord deformation index. Scale bar = 100 μm. (**E**) Cartoon depicting how the notochord deformation index was calculated. Normal vertebral growth should have a notochord deformation index of ~−1. (**F**) Notochord deformation index in 6.5–7.55 mm WT (blue) and 28 dpf *spzl*^-/- larvae (orange). Fish were matched by bone development. n = 3 for WT, n = 5 for *spzl*^-/-. Error bars are SD. The larger deviation illustrates that lack of symmetry in the notochord deformation in the *spzl*^-/- mutants. Black arrows point to the vertebral location of spine kinks and the magnitude (in μm) they deviate from the center line in *spzl*^-/- mutants (n = 18). Note that most kinks occur adjacent to areas where the asymmetric notochord deformation changes in orientation. (**G**) μCT image of a 2 mpf WT vertebra. Orange arrow indicates location of canal diameter measurements. (**H**) Canal diameter measurements from vertebrae 2–10 for WT (blue) and *spzl*^-/-(orange). The decreased size of the vertebral canal suggests the notochord is more deformable in *spzl*^-/- respect to WT. For WT n = 5. For *spzl* mutants n = 5.

The online version of this article includes the following figure supplement(s) for figure 9:

**Figure supplement 1.** Pronephros development occurs normally in *spzl* mutants.

model. Alternatively, the function of Dstyk may not be conserved across vertebrates. Nevertheless, it is interesting to note that some human patients with seemingly pathological variants in *DSTYK* also presented scoliosis (*Lee et al., 2017*). Determining whether mutations in this gene lead to CS in humans will require a dedicated study.

In conclusion, this work shows that notochord vacuoles play a crucial role in spine formation during vertebral bone growth.

# Materials and methods

## Fish stocks

Zebrafish (*Danio rerio*) were used in accordance with Duke University Institutional Animal Care and Use Committee (IACUC) guidelines. Zebrafish (Danio rerio) stocks were maintained at 28°C and bred as previously described (*Westerfield, 2000*). Zebrafish were raised in a circulating aquarium in tanks housing 1–10 fish/L. Genotype of zebrafish is specified in each figure legend. Male and female breeders from 3 to 9 months of age were used to generate fish for all experiments. 5–6 dpf zebrafish larvae from the Ekkwill (EK) or AB/TL background were used in this study. Strains generated for this study: *dstyk*^pd1133^; *spzl*^s739^. The *spzl*^s739^ allele was identified in an unrelated ENU based forward genetic screen that will be published elsewhere. Briefly, males were mutagenized using ENU and mated to WT females, F1 fish were then in-crossed to generate F2 families. Pairs from F2 families were then mated and F3 fish were screened for various recessive phenotypes. *spzl*^s739^ was identified serendipitously as an adult viable mutation causing shortening and kinking of the AP axis.

Previously published strains: *Tg(rcn3:gal4)pd* 1023, *Tg(UAS:GFP-rab32a; cmlc2:GFP)pd1083*, *Tg(rcn3:GFPrab32a)pd1153* (*Ellis et al., 2013a*); *Gt(Gal4FF)nksagff214a* (*Yamamoto et al., 2010*); *TgBAC(entpd5a:pkRED), Tg(nfatc-gal4)* and *Tg(col9a2:GFPCaaX)pd115* (*Garcia et al., 2017*); *Tg(col8a1a:GFPCaaX)pd pd1152* (*Garcia et al., 2017*); *Tg(osx:mcherry-NTR)pd43* (*Singh et al., 2012*); *nob*^hu3718^ (*Huitema et al., 2012*).

## DNA constructs

All constructs used for mosaic expression were generated using the same cloning strategies as before (*Garcia et al., 2017*). The kinase dead, *rcn3:dstykD757N:p2aGFP,* construct was made by using the Q5 site directed mutagenesis kit (NEB).

## Exome sequencing and mapping

DNA extraction, exome capture, sequencing, and positional cloning were performed exactly as we described previously (*Ryan et al., 2013*). Data were analyzed using SNPtrack (*Leshchiner et al., 2012*). The number of animals analyzed and recombinants observed from positional mapping are presented in *Figure 3A*.

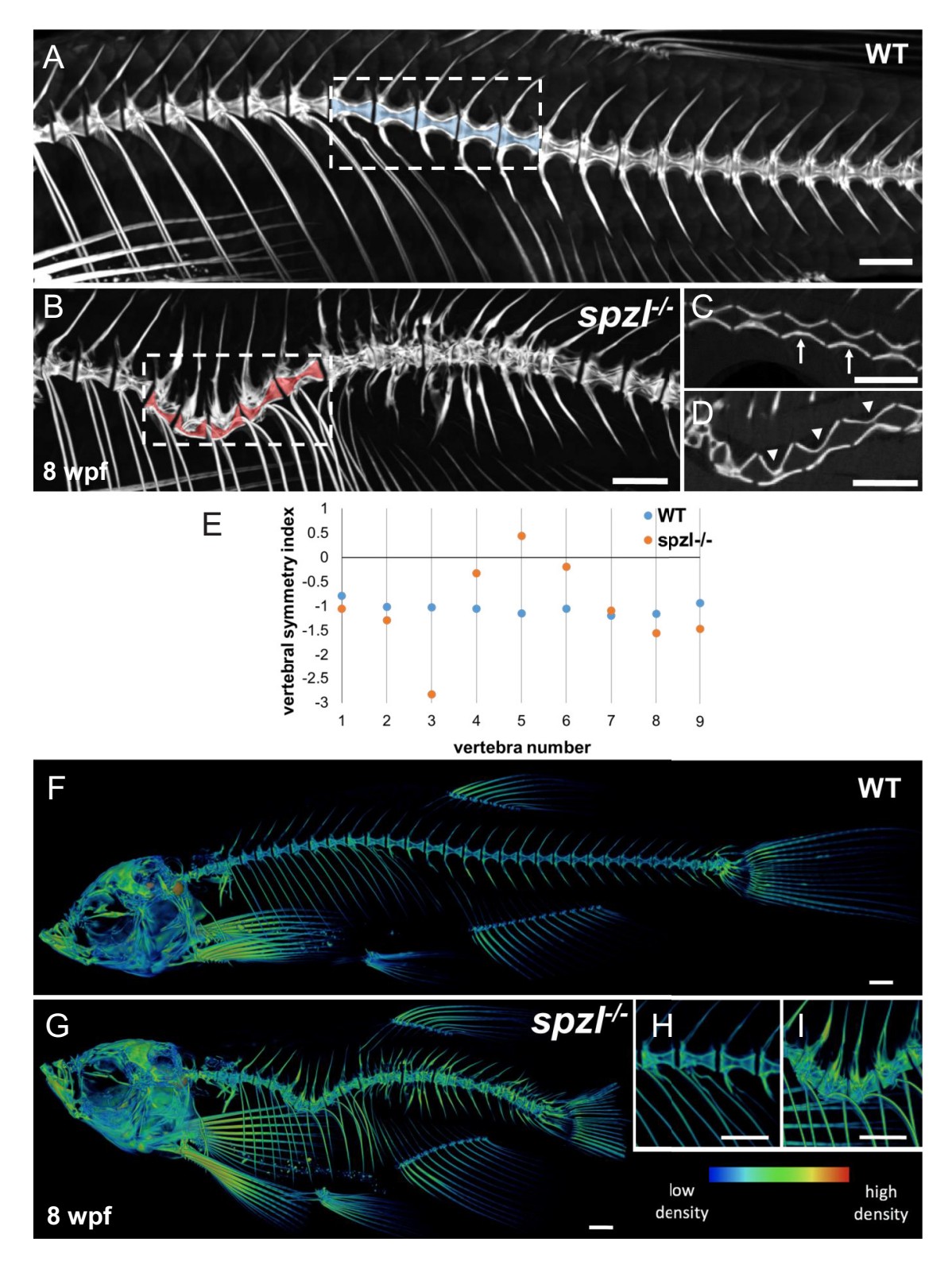

**Figure 10.** Vertebral malformations and increased mineralization in the spine of *spzl*-/- fish. (A–B) µCT images of the spine of WT (A) and *spzl*-/- (B) fish at 8 wpf. Vertebral canals are highlighted in blue (WT) and red (*spzl*-/-). (C) Optical section of the dotted box in (A) for WT. Arrows point to symmetrical vertebrae. (D) Optical section of the dotted box in (B). Arrowheads point to the vertebral asymmetry in *spzl*-/- associated with the spine kink. (E) Vertebral symmetry index, calculated as in **Figure 6E**, of WT and *spzl*-/- obtained from the spine kinking area. Numbers correspond to vertebra number

*Figure 10 continued on next page*

*Figure 10 continued*
counted from the first rib bearing vertebra. (F–G) μCT reconstruction of 6wpf WT and *spzl*<sup>-/-</sup> mutant zebrafish. The reconstructions were pseudo-colored to show relative bone density. (H–I) Magnified insets of the spine for WT and *spzl*<sup>-/-</sup>. Scale bar = 1 mm.
The online version of this article includes the following figure supplement(s) for figure 10:

**Figure supplement 1.** *dstyk* expression is not detectable at late stages in the notochord or in osteoblasts surrounding the notochord.

### In situ hybridization

To make in situ probes for *dstyk*, dstyk was amplified from cDNA using the following primers: dstyk_probe_F, 5' TGATGTCCGGCAGTATCGTG 3'; dstyk_probe_R, 5' TAATACGACTCACTA TAGGGCCATCTAAAAAGGAGCGTCGG 3'. Using the T7 promoter sequence included in the reverse primer, the PCR products were subjected to in vitro transcription using T7 RNA Polymerase (NEB). All in vitro transcription reactions were performed with DIG RNA Labeling Mix (Roche) to make digoxygenin-labeled RNA. In situ hybridization for dstyk was performed as described previously (*Navis and Bagnat, 2015*). Briefly, EK embryos were fixed in 4% PFA, dehydrated in 100% MeOH, and rehydrated. Then samples were digested with Proteinase K, hybridized with in situ probes, washed multiple times, incubated with secondary antibody, and stained using BM-Purple (Roche). Images were acquired on a Discovery V20 stereoscope (Zeiss).

### Genome editing and genotyping

Mutant lines were generated using CRISPR/Cas9. *dstyk* mutants were generated using CRISPRs targeting exon 7. Guide RNAs targeted sequences; dstyk-1–5' GGTCAATCCTCCAGCCATCA 3', dstyk-2–5' GGCGGCGCTCAGGCTCTCGA. Zebrafish embryos were injected at the one-cell stage with 50 pg total of guide RNA. Genotyping for both *dstyk* CRISPRs were performed using primers: forward, CCCGCTCATCTGATTACCCA; reverse, GGAGTGTGTGACCTGTCGTA. *spzl*<sup>-/-</sup> fish were genotyped using the same *dstyk* genotyping primers listed above. PCR products were then digested using the btgzI enzyme to distinguish WT and mutant bands.

### RNA isolation and reverse transcription PCR (RT-PCR)

RNA was extracted using the RNeasy Micro Kit (Qiagen) according to the manufacturer's protocol. cDNA was synthesized using Superscript III reverse transcriptase (Invitrogen). To generate cDNA for RT-PCR for *dstyk* the polydT primer was used for reverse transcription cDNA synthesis. Primers used for RT-PCR were; *dstyk*_RT_F ATGGAGAACCCGCAGAAGCCCCGG, and *dstyk*_RT_R TCAG TTGGAGTCCTCCAGGC.

### RA treatment

A 10 mM stock solution was made with all-trans RA (Sigma) and prepared in DMSO as described in *Laue et al. (2008)*; *Spoorendonk et al. (2008)*. At 14 dpf, 10 fish expressing *nfatc:gal4;UAS:mcherry* were put in 1L of fish water with 200 nM RA, 100 nM RA, and 0.0002% DMSO control, each in triplicate. Tanks were kept in the dark and the water and RA dose was changed daily. After 1 week of treatment, the surviving fish were stained with calcein, mounted and imaged as described below.

### Microscopy

Whole-mount confocal live imaging was performed on the Fluoview FV3000 (Olympus) with 30×/ 1.05 silicone oil objective (Olympus) and Fluoview software (Olympus). Fish were mounted on to glass bottom dishes in a 1.5% agarose mixture of egg water and 0.02% tricaine.

We also used an SP5 upright confocal microscope (Leica), using a 20x/0.70 HC PL APO oil objective (Leica) and Application Suite software (Leica). Fish were mounted on to slides with a small pool of 3% methylcellulose and 1X tricaine, encircled in a ring of vacuum grease. Coverslips were then placed on top to allow for imaging. We utilized the DIC images were taken with an Axio Imager.M1 microscope with 10 ×/0.3 EC Plan-NeoFluar objective, an AxioCamMRm camera, and AxioVision software (all from Carl Zeiss). Whole-mount imaging for body length measurements and skeletal preps were performed on a Zeiss Stereo Discovery.V20 microscope with 1.0x Achromat S FWD 63 mm objective (Carl Zeiss).

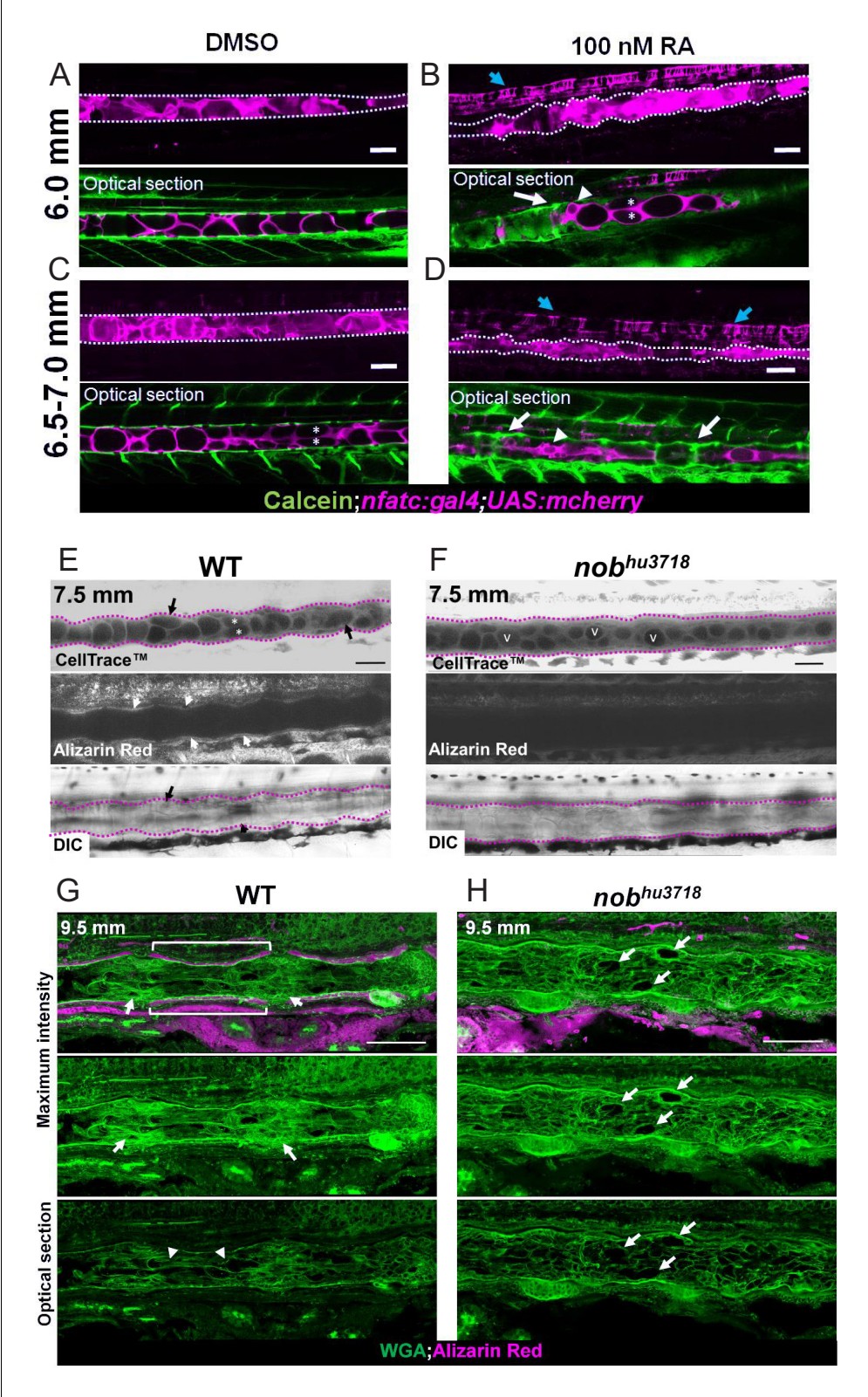

**Figure 11.** Notochord vacuole morphology and arrangement is altered with increasing or decreasing mineralization. (**A, C**) Live confocal images of fish treated with 0.0002% DMSO for 1 week at 6.0 mm and 6.5–7 mm respectively. (**B, D**) Live confocal images of fish treated with 100 nM RA for 1 week at 6.0 mm and 6.5–7 mm respectively. Fish expressed *nfatc:gal4;UAS:mcherry* in the notochord and were stained with calcein to label centra. Scale bar = 100 μm. n = 3 for DMSO control and RA treatment. Notochord is outlined with a dotted white line. White arrows point to areas of increased

*Figure 11 continued on next page*

*Figure 11 continued*

mineralization, blue arrows point to the floor plate. Arrowheads indicate areas of increased vacuole fragmentation. Asterisks mark vacuolated cell stacking. (**E**) Live confocal image of a 7.5 mm WT zebrafish stained with Cell Trace to visualize internal membranes and live alizarin to label mineralized bone, n = 3. (**F**) Live confocal image of a 7.5 mm *nob*^hu3718 stained with Cell Trace and alizarin red to label bone. n = 3. Black arrows point to vacuolated cells undergoing shape change. Asterisks mark stacked vacuolated cells. White arrows point to centra stained with alizarin red. v indicates rounded vacuoles that have not undergone shape change, fragmentation, or stacking. The notochord is outlined with a dotted fuchsia line. Scale bar = 100 μm. (**G–H**) Cryosections of a 9.5 mm WT (**G**) and *nob*^hu3718 (**H**) larvae stained with WGA (green) and alizarin red (magenta). Brackets indicate IVD domains. Arrows point to mineralized centra. The notochord is outlined with a dotted white line. Arrowheads point to mesenchymal condensations. Scale bar = 100 μm.

Whole-mount imaging for body length measurements and skeletal preps were performed on a Zeiss Stereo Discovery.V20 microscope with 1.0x Achromat S FWD 63 mm objective (Carl Zeiss).

Calcein imaging was done using an AX10 Zoom V116 Zeiss microscope equipped with a Plan Neofluar Z 1x objective and Zen software (Carl Zeiss). Fish were mounted on small plastic dishes in a mixture of 3% methylcellulose and 1X tricaine.

Light sheet microscopy was done using a Lightsheet Z.1 detection optics 20×/1.0 (water immersion) (Carl Zeiss). Fish were mounted in low melt agarose.

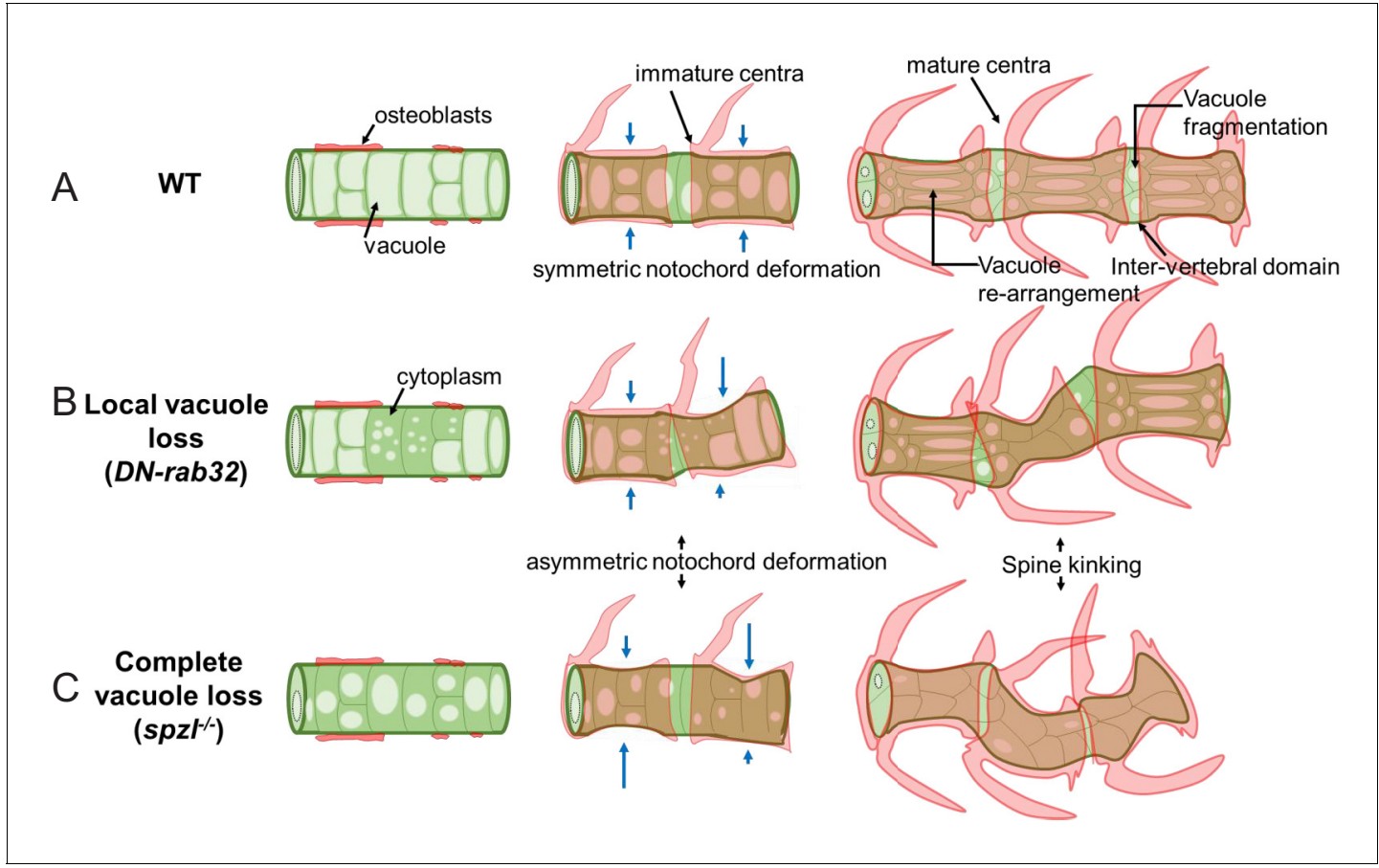

**Figure 12.** Working model depicting the proposed mechanism of notochord vacuole function. (**A**) Schematic depicting vacuole dynamics during spine formation in WT. Large and fully inflated vacuoles (vacuole lumen in light green) can rearrange under the growing centra and fragment in the inter-vertebral domain to absorb locally the compression generated by bone growth. (**B**) Schematic depicting localized loss of vacuoles driven by mosaic genetic manipulation. Vacuoles are lost in some cells in a confined area, leaving an increased amount of cytoplasm (dark green). Vacuole loss renders the notochord more deformable, causing asymmetric centra growth and spine kinking. Normal vacuole behavior around the affected area insulates the defect to one are. (**C**) Schematic depicting the effect of global loss of vacuoles in *spzl* mutants. Vacuoles are smaller and are progressively lost prior to centra development. Compressive centra growth can no longer be absorbed locally, leading to loss of compartmentalization and the development of spine kinks at various points of the axis. Blue arrows indicate degree of notochord deformations.

When necessary, images were minimally processed in ImageJ software (National Institutes of Health) for brightness. Image stitching was done in Fluoview software (Olympus).

## Image processing and 3D renderings

For all reconstructions, images were taken using a vacuolated cell specific line which labels the plasma membrane of the vacuolated cells (*col8a1a:GFPCaaX*). Images were acquired and stitched as described above. All renderings and reconstructions were made using Imaris v. 9.0.0 and 9.1.2 software (Bitplane USA, Concord, MA) as described previously (*Norman et al., 2018*). Notochord renderings were created using both the *surface* and *cells* features (Imaris software, v. 9.0.0 and 9.1.2, respectively). Volume and sphericity measurements were also taken from these renderings.

Cell arrangement patterns were determined as previously published (*Norman et al., 2018*).

To determine nuclear shapes, 14 dpf larvae expressing *SAG:gal4;UAS:GFP* were fixed in 4% PFA overnight and sectioned as previously described (*Bagnat et al., 2010*). Sections were then stained with DAPI (Vector Laboratiories) and mounted using Fluoromount-G (Southern Biotech). Imaris was used as described above to reconstruct nuclei expressing both the vacuole marker and DAPI. Volume and sphericity measurements were also determined as described above.

The sphericity of vacuolated cells was determined by using the ratio of the surface area of a sphere with the same volume as each given cell to the surface area of that cell. Cell volume and cell surface area values were obtained using the cells feature in Imaris. Cell volumes were used to calculate the radius of a sphere of equal volume and then the radius of sphere of equal volume was used to calculate the corresponding surface area. The sphericity of the vacuolated cell nuclei was generated by the surfaces feature in Imaris.

## Vital stains and histological methods

The vital dye Cell Trace (Molecular Probes) and calcein (Sigma) were used for some live imaging as described before (*Ellis et al., 2013a*).

For in vivo alizarin red staining, a 0.01% ARS solution was prepared in egg water as previously described (*Bensimon-Brito et al., 2016*). Zebrafish were submerged in the solution for 15 min prior to imaging or fixation.

For skeletal preparations, zebrafish between 21 dpf-2 mpf were fixed in 4% PFA and eviscerated. They were then stained with alizarin red as previously described *Ellis et al. (2013a)*, and treated with 1% KOH to clear tissue from the bone. Once the skeleton was clear of tissue, the spine was imaged as indicated above.

## Cryosectioning and staining with wheat germ agglutinin (WGA)

Late stage WT and *nob^hu3718* larvae (7.5 mm-9 mm SL) were stained with live alizarin red and fixed in 4% PFA. Fish were then cryopreserved in 30% sucrose and embedded in OCT for cryo-sectioning. 12 μm sagittal sections were collected and subsequently stained with DAPI and WGA Alexa Fluor 488. Sections were then mounted with Fluoromount-G (SouthernBiotech) and imaged using a Leica SP8 inverted confocal microscope.

## Micro-computed tomography (μCT) analysis

Samples were fixed with paraformaldehyde and placed inside a low-density foam mold. Scanning was performed on a SkyScan 1275 high resolution μCT system (Bruker, Kontich, Belgium) with an x-ray source voltage and current of 50 kV and 200 μA respectively. Over 2800 projection images were generated over 360° with a 0.1° rotation step and 6 averaging frames, isotropic resolution was 10.5 μm. Thresholding, ring artifact reduction, and beam hardening corrections were consistent across on all scans during reconstruction using NRecon (Bruker, Kontich, Belgium). Reconstructed BMP slices were analyzed using Amira 6.2 (Thermo Fisher Scientific FEI, Hillsboro, Oregon, USA). Density heatmaps were generated with the volume rendering module and physics load transfer function, canal measurements were taken with the 2D measurement tool, and optical sections were generated with the orthoslice module.

## Acknowledgements

We would like to thank Jieun Park and Kelsey Oonk for help with ISH, the Duke Zebrafish Core Facility for fish care, and the Duke Light Imaging facility for assistance with imaging and data processing (S10 OD020010 - Lightsheet Imaging System). We would like to thank Anne Marie Pendergast for advising the design of kinase mutants. We thank Boris Shraiman and Francis Corson for discussions, Curtis Boswell for insights into spine defects, and Ken Poss, Alessandro De Simone and Noah Mitchell for critical reading of our manuscript.

This work was funded by NIH grant R01AR065439 and a HHMI Faculty Scholars Award to MB, R01HL54737 to DYRS, and R00GM105874 and R03HD091634 to SKM.

## Additional information

### Competing interests

Didier YR Stainier: Senior editor, *eLife*. The other authors declare that no competing interests exist.

### Funding

| Funder | Grant reference number | Author |
| --- | --- | --- |
| National Institutes of Health | R01AR065439 | Michel Bagnat |
| Howard Hughes Medical Institute | Faculty Scholars | Michel Bagnat |
| National Institutes of Health | R01HL54737 | Didier YR Stainier |
| National Institutes of Health | R00GM105874 | Sarah K McMenamin |
| National Institutes of Health | R03HD091634 | Sarah K McMenamin |

The funders had no role in study design, data collection and interpretation, or the decision to submit the work for publication.

### Author contributions

Jennifer Bagwell, Conceptualization, Data curation, Formal analysis, Investigation, Visualization, Project administration; James Norman, Investigation, Data curation, Formal analysis, visualization; Kathryn Ellis, Data curation, Formal analysis, Investigation; Brianna Peskin, James Hwang, Xiaoyan Ge, Stacy V Nguyen, Investigation; Sarah K McMenamin, Formal analysis, Investigation; Didier YR Stainier, Resources; Michel Bagnat, Conceptualization, Resources, Data curation, Supervision, Funding acquisition, Project administration

### Author ORCIDs

Jennifer Bagwell (ID) https://orcid.org/0000-0002-8873-0897
Stacy V Nguyen (ID) https://orcid.org/0000-0002-2641-3984
Sarah K McMenamin (ID) https://orcid.org/0000-0002-1154-5810
Didier YR Stainier (ID) http://orcid.org/0000-0002-0382-0026
Michel Bagnat (ID) https://orcid.org/0000-0002-3829-0168

### Ethics

Animal experimentation: Zebrafish (*Danio rerio*) were used in accordance with Duke University Institutional Animal Care and Use Committee (IACUC) guidelines and approved under our animal protocol A089-17-04.

### Decision letter and Author response

Decision letter https://doi.org/10.7554/eLife.51221.sa1
Author response https://doi.org/10.7554/eLife.51221.sa2

## Additional files

### Supplementary files
- Transparent reporting form
- Source data 1. Raw data for all measurements, statistics, and graphs.

### Data availability

All data generated or analyses during this study are included in the manuscript and supporting files. Source data files have been provided as indicated. Data has been deposited to Dryad, under the DOI: 10.5061/dryad.73n5tb2tb. Due to their large size, raw image files can be accessed upon request.

The following dataset was generated:

| Author(s) | Year | Dataset title | Dataset URL | Database and Identifier |
|---|---|---|---|---|
| Jennifer Bagwell, James Hwang, Kathryn L. Ellis, Brianna Peskin, James Norman, Xiaoyan Ge, Stacy Nguyen, Sarah K. McMenamin, Didier Y. Stainier, Michel Bagnat. | 2020 | Data from: Notochord vacuoles absorb compressive bone growth during zebrafish spine formation | http://doi.org/10.5061/dryad.73n5tb2tb | Dryad Digital Repository, 10.5061/dryad.73n5tb2tb |

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
