## [Decision Letter]

**Acceptance summary:**

This paper investigates the role of notochordal vacuoles in formation of the spinal column. Using an unbiased forward genetic approach in zebrafish, the authors identify a mutation (*spzl*) that is associated with vacuole fragmentation and discrete spinal kinking. They localize this mutation to the *dstyk* gene and show that the *spz*l mutants exhibit impaired vacuole fusion due to loss of *dstyk* activity. Notochordal vacuoles are proposed to regulate spine morphogenesis by resisting compressive forces during vertebral bone growth. Specifically, the vacuoles are proposed to reorientate to accommodate symmetric formation of the vertebral centra. This work offers important new insights into the critical role of notochordal vacuoles in the mechanical regulation of spinal column formation. Results suggest that clinically, conditions associated with vacuole fragmentation may result in congenital spinal deformity.

**Decision letter after peer review:**

Thank you for submitting your article "Notochord vacuoles absorb compressive bone growth during zebrafish spine formation" for consideration by *eLife*. Your article has been reviewed by two peer reviewers, and the evaluation has been overseen by a Reviewing Editor, Lila Solnica-Krezel, who also read the manuscript. Richard White served as the Senior Editor. The reviewers have opted to remain anonymous.

The reviewers have discussed the reviews with one another and the Reviewing Editor has drafted this decision to help you prepare a revised submission.

Summary:

There were different perspectives on the novelty and significance of the findings. One view is that this work presents important new insights into the critical role of notochordal vacuoles in the mechanical regulation of spinal column formation. Results suggest that clinically, conditions associated with vacuole fragmentation may result in congenital spinal deformity. A different perspective is that after Ellis et al., 2013 paper from these authors, providing the first link between defects in the notochord vacuoles, spine skeleton and scoliosis, the novelty of the findings presented in this manuscript is rather limited. In particular, as no data on the molecular mechanisms underlying the role of Dstyk during vacuole formation or maintenance are provided.

Overall, the manuscript is very well written, the data are of highest technical quality, the images are beautiful and the conclusions are sound. In addition, the findings are very interesting, but not entirely novel. Moreover, the key thesis of the manuscript that *dstyk* functions in the notochord to regulate vacuole morphogenesis and their ability to resist compressive forces during bone growth, requires further experimental support. Experimentally addressing this question as well as the role of vacuole fragmentation within intervertebral regions and testing function of maternally contributed *dstyk* function, would significantly strengthen the manuscript. The manuscript has a very good chance to improve significantly even within two months of revisions, when analyses on the intrinsic role of the notochord during spine formation are extended.

Essential revisions:

1) The authors interpret the spine phenotype of *dstyk* mutants as evidence for an essential role of notochord cells for proper spine formation and to avoid scoliosis, contrasting other scoliosis disease genes that act in the paraxial mesoderm. However, while this interpretation is most likely correct, zebrafish *dstyk* is also expressed in other tissues, while mutations in human have been implicated with kidney agenesis. In this light, it should be more rigorously shown that it is indeed the loss of Dstyk function in the notochord that leads to the spine defects.

The authors should extend their in situ hybridization analyses to show how long Dstyk is expressed in the notochord and whether or whether not it is also expressed in paraxial mesoderm and osteoblasts.

Furthermore, the analyses on the notochord-specific requirement of Dstyk should be extended. The rescue experiments via injection of DNA constructs driving expression of labelled wild-type under the control of the rcn3 promoter provide first evidence. However, at least the rescue shown for the representative example in Figure 2L,M is not very convincing (isn't the indicated vacuole far too large?). Also, it is not clear whether the transgene was only expressed in notochordal cells (the endogenous rcn3 gene also shows other expression domains, e.g. in the floorplate and the somites). This needs to be clearly stated. If the transgene was not exclusively expressed in the notochord, it would be helpful to generate mosaics via cell or shield transplantation experiments to investigate whether a wild-type notochord in a mutant host leads to normal spine development and whether a mutant notochord in a wild-type host leads to spine defects as in regular mutants. Here, to more specifically target notochord cells, the authors might want to generate mosaics by transplanting wild-type cells directly into the shield of *dstyk* mutants and vice versa. These studies should definitively not take longer than two months.

2) Loss of zygotic Dstyp function does not affect the initial formation, but only the later maintenance of notochord vacuoles. In addition, it does not affect anterior, but only posterior regions of the spine. The authors propose that this might be due to a maternal supply of *dstyk* gene products that compensates for the zygotic loss. However, no such maternal expression or function is shown thus far, even though the Zebrafish Expression Atlas online tool https://www.ebi.ac.uk/gxa/experiments/E-ERAD-475/Results shows strong maternal *dstyk* expression. Therefore, the authors should add pre-MBT stages to their in situ hybridizations in Figure 2, complemented by RT-PCR analyses to demonstrate maternal *dstyk* expression. Furthermore, if homozygous females are fertile or can be successfully squeezed for in vitro fertilization, maternal-zygotic mutants should be generated and investigated for earlier defects in notochord vacuole formation.

3) The conclusion from Figure 7G that bone mineralization is mechanically regulated, based solely on apparent increased mineralization in mutants is too speculative. Have the authors explored whether *dstyk* is expressed in osteoblasts? For example, have the ISH experiments in Figure 2E been performed at later stages of development (after the onset of mineralization)? If so, this could present an additional/alternative explanation for the bone defects observed.

4) In light of the observed vacuole fragmentation of notochord cells within intervertebral regions (thus not exposed to mechanical pressure from the mineralized bone of the forming centra), it would be interesting to study vacuole shapes in fish with expanded centra mineralization (e.g. after treatment with retinoic acid) and in the absence of bone mineralization (e.g. in the no bone mutant). Unaltered vacuole fragmentation patterns under such conditions would strongly point to a notochord-intrinsic patterning mechanism and be of broad significance. Hence, the authors should extend on their observation that even in wild types, the vacuoles within the intervertebral spaces undergo fragmentation. Thus, it would be important to investigate whether this segmented vacuole fragmentation is an intrinsic feature of the notochord or whether it depends on mechanical forces resulting from the biomineralization of the centra – which in turn could be under the control of external regulators. For these studies, no complex genetic scenarios are required. The authors would for instance just need plain no bone / *entpd5a* mutants (stained with Boedipy TR methylester dye) or wild-type or *Tg(SAGFF214A:gal4); Tg(UAS:GFP)* fish treated with retinoic acid or DEAB and analyzed at a standard length of 8.7 mm (3-4 weeks of age). Thus, also these studies could be completed within two months.

Title: With respect to the title, it would be more appropriate to say the vacuoles "resist" rather than "absorb" compressive one growth.

[Editors' note: further revisions were suggested prior to acceptance, as described below.]

Thank you for resubmitting your work entitled "Notochord vacuoles absorb compressive bone growth during zebrafish spine formation" for further consideration by *eLife*. Your revised article has been evaluated by Richard White (Senior Editor) and a Reviewing Editor.

The manuscript has been improved and the authors largely addressed the questions and concerns of the reviewers. The revisions significantly strengthened the manuscripts. This work significantly advances our understanding of how the notochord and notochord vacuoles influence spine morphogenesis and identify an essential role of Dstyk/Spatzle kinase in vacuole fusion. However, there are some remaining issues that need to be addressed before acceptance, as outlined below:

1) This revised manuscript provides many lines of evidence to support the model whereby notochord vacuoles generate pressure and their size/shape underlies the notochord pressure required for proper spine formation. However, as the authors note, current methods do not allow for direct pressure measurements in the notochord. Therefore, many of the conclusions, including "…upon fragmentation or loss of notochord vacuoles, the notochord loses its internal pressure" are inferences. These conclusions should be accordingly phrased.

2) The authors state "In contrast, mutations affecting several different tissues can cause AIS; these tissues include the neural tube (Grimes et al., 2016; Hayes et al., 2014; Sternberg et al., 2018), cartilage (Karner et al., 2015), and paraxial mesoderm (Haller et al., 2018; Liu et al., 2017)." Liu et al., 2017 did not attribute the scoliosis phenotype in stat3 zebrafish mutants to its function in paraxial mesoderm. One possible mechanism proposed, was a correlation between the onset of a massive inflammation and scoliosis phenotype in these mutant fish.

3) Several figures are still not labeled with developmental stage information. Please add hpf or dpf to Figure 1 panels E,F,G, and whenever appropriate.

4) In revised Figure 3F please ensure that all allele numbers are in italics.

5) Please use capital letters: Zspzl and MZspzl to indicate zygotic and maternal zygotic adjectives in text and figures, otherwise these symbols merge with the gene name abbreviation and are confusing.

6) Figure 6 is beautiful and informative. This reviewer understands that proper staging at these later stages is provided in mm and the corresponding calendar time differs between fish facilities, but panels A-D show staging in mm and E,F in dpf. Could both staging measures be provided?

7) In addition, what is the genotype of animals shown in F? From the text one learns that this is localized overexpression of *DN-GFP-Rab32a*, this should be indicated in the figure.

8) "…and devises such as pressure gauges" – do the authors mean "devices"?

---

## [Author Response]

Summary:There were different perspectives on the novelty and significance of the findings. One view is that this work presents important new insights into the critical role of notochordal vacuoles in the mechanical regulation of spinal column formation. Results suggest that clinically, conditions associated with vacuole fragmentation may result in congenital spinal deformity. A different perspective is that after Ellis et al., 2013 paper from these authors, providing the first link between defects in the notochord vacuoles, spine skeleton and scoliosis, the novelty of the findings presented in this manuscript is rather limited. In particular, as no data on the molecular mechanisms underlying the role of Dstyk during vacuole formation or maintenance are provided.Overall, the manuscript is very well written, the data are of highest technical quality, the images are beautiful and the conclusions are sound. In addition, the findings are very interesting, but not entirely novel. Moreover, the key thesis of the manuscript that dstyk functions in the notochord to regulate vacuole morphogenesis and their ability to resist compressive forces during bone growth, requires further experimental support. Experimentally addressing this question as well as the role of vacuole fragmentation within intervertebral regions and testing function of maternally contributed dstyk function, would significantly strengthen the manuscript. The manuscript has a very good chance to improve significantly even within two months of revisions, when analyses on the intrinsic role of the notochord during spine formation are extended.

We thank the reviewers for their insightful comments and for their kind words praising the quality of our manuscript. We agree it was needed to further explore and substantiate some of the more mechanistic findings to strengthen our conclusions. We feel that the changes and additions we introduced have significantly improved our manuscript.

We would like to address here the issue of novelty by providing our perspective. We believe this work provides new mechanistic insight into the role of notochord vacuoles in spine formation that go well beyond what we showed previously in Ellis et al., 2013. Specifically, we show that loss of vacuole integrity leads to vertebral malformations during bone growth, and that vacuolated cells respond to compressive bone growth by stacking and fragmenting vacuoles to absorb bone growth. These are in our view novel mechanistic insights. With respect to Dstyk function, we show that the kinase activity regulates membrane fusion and that fusion of pre-vacuolar compartments (i.e. vacuolinos) is a rate limiting step. While we have not identified phosphorylation targets of Dstyk we were able to define its developmental function.

Essential revisions:1) The authors interpret the spine phenotype of dstyk mutants as evidence for an essential role of notochord cells for proper spine formation and to avoid scoliosis, contrasting other scoliosis disease genes that act in the paraxial mesoderm. However, while this interpretation is most likely correct, zebrafish dstyk is also expressed in other tissues, while mutations in human have been implicated with kidney agenesis. In this light, it should be more rigorously shown that it is indeed the loss of Dstyk function in the notochord that leads to the spine defects.The authors should extend their in situ hybridization analyses to show how long Dstyk is expressed in the notochord and whether or whether not it is also expressed in paraxial mesoderm and osteoblasts.

We can detect the *dstyk* transcript in the notochord by ISH until about 72 hpf. However, because vacuolated cells never divide after differentiation gene products therefore persist for a long time. This is why vacuole loss is progressive and it takes several days for the phenotype to fully manifest.

To investigate whether *dstyk* is expressed in the paraxial mesoderm and osteoblasts we performed ISH in cryosections at relevant stages and were not able to detect any trace of the *dstyk* transcript, while *osterix* expression in osteoblast was prominent. We have added a supplemental figure with these data.

Furthermore, the analyses on the notochord-specific requirement of Dstyk should be extended. The rescue experiments via injection of DNA constructs driving expression of labelled wild-type under the control of the rcn3 promoter provide first evidence. However, at least the rescue shown for the representative example in Figure 2L,M is not very convincing (isn't the indicated vacuole far too large?).

We thank the reviewers for pointing this out. Indeed, we consistently observe that cells overexpressing Dstyk have larger vacuoles. We also detected that when sheath cells overexpressed Dstyk they appeared to make vacuoles. To investigate this further we injected our construct into a line that carries a sheath cell marker and observed, as shown in the revised Figure 4, that they clearly make vacuoles, thus strengthening our conclusions. This is consistent with previous work showing that vacuolated cells can regenerate lost vacuolated cells and suggests Dstyk may be important for notochord regeneration, we plan to test this idea in future work.

Also, it is not clear whether the transgene was only expressed in notochordal cells (the endogenous rcn3 gene also shows other expression domains, e.g. in the floorplate and the somites). This needs to be clearly stated.

In our rescue experiments we have consistent expression in notochord cells only, sheath and vacuolated cells, and not in the floor plate. We only scored larvae in which cell autonomous effects are clear, i. e. single expressing cells not surrounding by other expressing cells. We are confident about these results and have added more examples to the figure, along with the sheath cell phenotype mentioned above.

If the transgene was not exclusively expressed in the notochord, it would be helpful to generate mosaics via cell or shield transplantation experiments to investigate whether a wild-type notochord in a mutant host leads to normal spine development and whether a mutant notochord in a wild-type host leads to spine defects as in regular mutants. Here, to more specifically target notochord cells, the authors might want to generate mosaics by transplanting wild-type cells directly into the shield of dstyk mutants and vice versa. These studies should definitively not take longer than two months.

We attempted this experiment several times, but we unfortunately were unable to generate viable embryos following transplants. Clearly, this is a traditional experimental skill that we need to learn from experts, nobody at Duke has expertise with cell transplantation. However, we feel this point is adequately addressed by our rescue and loss of function data.

2) Loss of zygotic Dstyp function does not affect the initial formation, but only the later maintenance of notochord vacuoles. In addition, it does not affect anterior, but only posterior regions of the spine. The authors propose that this might be due to a maternal supply of dstyk gene products that compensates for the zygotic loss. However, no such maternal expression or function is shown thus far, even though the Zebrafish Expression Atlas online tool https://www.ebi.ac.uk/gxa/experiments/E-ERAD-475/Results shows strong maternal dstyk expression. Therefore, the authors should add pre-MBT stages to their in situ hybridizations in Figure 2, complemented by RT-PCR analyses to demonstrate maternal dstyk expression. Furthermore, if homozygous females are fertile or can be successfully squeezed for in vitro fertilization, maternal-zygotic mutants should be generated and investigated for earlier defects in notochord vacuole formation.

We thank the reviewers for the suggestion. To answer this point, we performed ISH at the once-cell stage and detected robust expression of *dstyk* as expected, shown in a panel added to the corresponding figure, and also detect the transcript by RT-PCR (not shown). We also generated mz mutants and found they present very severe early phenotypes, perhaps pointing to a role in oogenesis. In embryos that survive through notochord morphogenesis we found that vacuolated cells differentiate, but do not make vacuoles. This result, our rescue experiments and our imaging data strongly support a role for *dstyk* in vacuole biogenesis. We have added a supplemental figure to illustrate the mz mutant phenotypes.

3) The conclusion from Figure 7G that bone mineralization is mechanically regulated, based solely on apparent increased mineralization in mutants is too speculative. Have the authors explored whether dstyk is expressed in osteoblasts? For example, have the ISH experiments in Figure 2E been performed at later stages of development (after the onset of mineralization)? If so, this could present an additional/alternative explanation for the bone defects observed.

We agree the statement was too speculative at that point of the manuscript and have modified it. To answer the specific question, we performed ISH (please see answer to point 1) and found *dstyk* is not seem to be expressed in osteoblasts.

4) In light of the observed vacuole fragmentation of notochord cells within intervertebral regions (thus not exposed to mechanical pressure from the mineralized bone of the forming centra), it would be interesting to study vacuole shapes in fish with expanded centra mineralization (e.g. after treatment with retinoic acid) and in the absence of bone mineralization (e.g. in the no bone mutant). Unaltered vacuole fragmentation patterns under such conditions would strongly point to a notochord-intrinsic patterning mechanism and be of broad significance. Hence, the authors should extend on their observation that even in wild types, the vacuoles within the intervertebral spaces undergo fragmentation. Thus, it would be important to investigate whether this segmented vacuole fragmentation is an intrinsic feature of the notochord or whether it depends on mechanical forces resulting from the biomineralization of the centra – which in turn could be under the control of external regulators. For these studies, no complex genetic scenarios are required. The authors would for instance just need plain no bone / entpd5a mutants (stained with Boedipy TR methylester dye) or wild-type or Tg(SAGFF214A:gal4); Tg(UAS:GFP) fish treated with retinoic acid or DEAB and analyzed at a standard length of 8.7 mm (3-4 weeks of age). Thus, also these studies could be completed within two months.

We thank the reviewers for these insightful suggestions. We carried out the suggested experiments as well as others that complement and extend those and other related observations.

We have added new data with a thorough quantitative characterization of WT and mutant vacuolated cells including their arrangement and morphological features that help explain the axis elongation defects. We also wanted to somehow determine whether vacuole fragmentation leads to a loss of internal notochord pressure and the results are striking. We figured out that as a consequence of vacuole inflation the vacuolated cell nuclei become displaced to the periphery of the notochord core and are deformed into bent disks. By contrast, in *spzl* mutants the nuclei are uniformly distributed throughout the notochord core and the vacuolated cell nuclei are more spherical. These data have been added to the new Figure 8.

To analyze the effect of notochord compression on vacuolated cell shape and arrangement we monitored quantitatively how cell shapes change over time using 3D renderings. We found that first with chordacentra formation and then with bone growth there is a significant change in morphology that is characterized by the emergence of highly elliptical cells, i.e. the cells that are stacked underneath centra. We have added these data to Figure 6.

To determine the effect of mineralization on vacuolated cells we monitored the notochord in *nob* mutants as suggested. We had heterozygous fish in the lab, but without the transgenes in the background, which made the analyses more difficult. We used the Bodipy TR methylester (Cell Trace) dye as far into development as it is possible and then switched to cryosections and WGA stainings. Initially, there is some level of compression from mesenchymal condensations, but these do not progress and as a result the early stages of vacuolated cell stacking are not apparent. Then, once bone mineralization has progressed in WT a clear distinction in vacuolated cell arrangements can be seen, with IVDs clearly bulging and vacuolated cells packing tightly. This reorganization of vacuolated cells does not occur in *nob* mutants, which retain the early arrangement. We would have liked to monitor this process at higher resolution, but to incorporate proper markers we would have needed another 3-4 months as late expression requires two separate transgenes. We would also like to add that the process of differentiation that leads to IVD formation is complex and goes beyond the developmental window we covered here. Cleary, future work will be needed to fully understand this process and will involve the development of new experimental methods and even theoretical approaches as this is a complex soft matter problem for which there is no theoretical framework.

We also performed the RA acid experiment as suggested. Following previously published protocols we were able to reproduce the increased mineralization effect of exogenous RA. We observed that as mineralization proceeds much further and at a faster rate upon RA treatment vacuoles stack prematurely and then are obliterated as the notochord is compressed further. These data have been included in the new Figure 11.

In addition to the results summarized above, we were also able to increase our numbers on the microCT analyses, further strengthening our conclusions.

Together, the results shown in the new Figure 11 add additional support to our initial conclusions. Future work should explore the mechanisms that mediate the effect of compression on vacuolated cell arrangement and vacuole integrity, we think this might be a mechanically driven self-organization process and is thus of great interest for us.

Title: With respect to the title, it would be more appropriate to say the vacuoles "resist" rather than "absorb" compressive one growth.

We think that both are appropriate and take place at different times, perhaps “resist and absorb” is the most accurate formulation, but would like to keep “absorb” as it reflects an important insight.

[Editors' note: further revisions were suggested prior to acceptance, as described below.]

The manuscript has been improved and the authors largely addressed the questions and concerns of the reviewers. The revisions significantly strengthened the manuscripts. This work significantly advances our understanding of how the notochord and notochord vacuoles influence spine morphogenesis and identify an essential role of Dstyk/Spatzle kinase in vacuole fusion. However, there are some remaining issues that need to be addressed before acceptance, as outlined below:1) This revised manuscript provides many lines of evidence to support the model whereby notochord vacuoles generate pressure and their size/shape underlies the notochord pressure required for proper spine formation. However, as the authors note, current methods do not allow for direct pressure measurements in the notochord. Therefore, many of the conclusions, including "…upon fragmentation or loss of notochord vacuoles, the notochord loses its internal pressure" are inferences. These conclusions should be accordingly phrased.

We agree with the editors and have edited those statements as suggested. We used wording such as “presents a phenotype consistent with reduced internal pressure”, “the notochord rod appears to lose its internal pressure”, and “rod that seems to have low internal pressure”. However, we left the title as it was as it would otherwise be difficult to express the message in a compact way.

2) The authors state "In contrast, mutations affecting several different tissues can cause AIS; these tissues include the neural tube (Grimes et al., 2016; Hayes et al., 2014; Sternberg et al., 2018), cartilage (Karner et al., 2015), and paraxial mesoderm (Haller et al., 2018; Liu et al., 2017)." Liu et al., 2017 did not attribute the scoliosis phenotype in stat3 zebrafish mutants to its function in paraxial mesoderm. One possible mechanism proposed, was a correlation between the onset of a massive inflammation and scoliosis phenotype in these mutant fish.

We thank the editors for pointing out our mistake. We have re-phrased our mention of the work of Liu et al. as suggested.

3) Several figures are still not labeled with developmental stage information. Please add hpf or dpf to Figure 1 panels E,F,G, and whenever appropriate.

We have added the missing information.

4) In revised Figure 3F please ensure that all allele numbers are in italics.

Done.

5) Please use capital letters: Zspzl and MZspzl to indicate zygotic and maternal zygotic adjectives in text and figures, otherwise these symbols merge with the gene name abbreviation and are confusing.

We agree and have changed the labeling as suggested.

6) Figure 6 is beautiful and informative. This reviewer understands that proper staging at these later stages is provided in mm and the corresponding calendar time differs between fish facilities, but panels A-D show staging in mm and E,F in dpf. Could both staging measures be provided?

For panel E we have replaced staging in dpf by standard length as suggested. However, for panel F we need to leave the age in dpf because our manipulation causes kinking of the axis, which precludes accurate staging by length.

7) In addition, what is the genotype of animals shown in F? From the text one learns that this is localized overexpression of DN-GFP-Rab32a, this should be indicated in the figure.

Those are WT fish expressing the transgenes indicated in the figure. We have modified the label to indicate the expression of *DN-GFP-Rab32a*.

8) "…and devises such as pressure gauges" – do the authors mean "devices"?

Yes, thank you for pointing out that typo.